# The RNF168 paralog RNF169 defines a new class of ubiquitylated histone reader involved in the response to DNA damage

Julianne Kitevski-LeBlanc[1,2,3†], Amélie Fradet-Turcotte[4,5†], Predrag Kukic[6‡], Marcus D Wilson[4‡§], Guillem Portella[6], Tairan Yuwen[1,2,3], Stephanie Panier[1,4§], Shili Duan[7,8], Marella D Canny[9], Hugo van Ingen[10], Cheryl H Arrowsmith[7,8,9], John L Rubinstein[1,9,11], Michele Vendruscolo[6], Daniel Durocher[1,4*], Lewis E Kay[1,2,3,11*]

[1]Department of Molecular Genetics, University of Toronto, Toronto, Canada; [2]Department of Biochemistry, University of Toronto, Toronto, Canada; [3]Department of Chemistry, University of Toronto, Toronto, Canada; [4]The Lunenfeld-Tanenbaum Research Institute, Mount Sinai Hospital, Toronto, Canada; [5]Laval University Cancer Research Center, Oncology Axis – Centre Hospitalier Universitaire de Québec Research Center – Université Laval, Hôtel-Dieu de Québec, Québec City, Canada; [6]Department of Chemistry, University of Cambridge, Cambridge, United Kingdom; [7]Structural Genomics Consortium, University of Toronto, Toronto, Canada; [8]Princess Margret Cancer Centre, Toronto, Canada; [9]Department of Medical Biophysics, University of Toronto, Toronto, Canada; [10]Macromolecular Biochemistry, Leiden Institute of Chemistry, Leiden University, Leiden, The Netherlands; [11]Molecular Structure and Function Program, The Hospital for Sick Children Research Institute, Toronto, Canada

*For correspondence: durocher@lunenfeld.ca (DD); kay@pound.med.utoronto.ca (LEK)

†These authors contributed equally to this work
‡These authors also contributed equally to this work

Present address: §The Francis Crick Research Institute, London, United kingdom

Competing interests: The authors declare that no competing interests exist.

**Abstract** Site-specific histone ubiquitylation plays a central role in orchestrating the response to DNA double-strand breaks (DSBs). DSBs elicit a cascade of events controlled by the ubiquitin ligase RNF168, which promotes the accumulation of repair factors such as 53BP1 and BRCA1 on the chromatin flanking the break site. RNF168 also promotes its own accumulation, and that of its paralog RNF169, but how they recognize ubiquitylated chromatin is unknown. Using methyl-TROSY solution NMR spectroscopy and molecular dynamics simulations, we present an atomic resolution model of human RNF169 binding to a ubiquitylated nucleosome, and validate it by electron cryomicroscopy. We establish that RNF169 binds to ubiquitylated H2A-Lys13/Lys15 in a manner that involves its canonical ubiquitin-binding helix and a pair of arginine-rich motifs that interact with the nucleosome acidic patch. This three-pronged interaction mechanism is distinct from that by which 53BP1 binds to ubiquitylated H2A-Lys15 highlighting the diversity in site-specific recognition of ubiquitylated nucleosomes.

## Introduction

Protein ubiquitylation is a versatile signal that controls a wide range of cellular functions including protein degradation, transcription, immunity, inflammation, endocytosis and DNA repair (*Komander and Rape, 2012*; *Jackson and Durocher, 2013*). The covalent attachment of ubiquitin (ub), a 76 residue polypeptide, to a target protein is achieved by the sequential activity of ubiquitin-activating (E1), ubiquitin-conjugating (E2) and ubiquitin-ligating (E3) enzymes resulting in the

**eLife digest** Inside cells, genetic information is encoded by molecules of DNA. It is important for a cell to quickly identify and repair any damage to DNA to prevent harmful changes in the genetic information. In humans and other animals failures in DNA repair can lead to cancer and other diseases.

A molecule of DNA is made of two strands that twist together to form a double helix. Most of the DNA in an animal cell is organised by proteins called histones. Groups of eight histones are wrapped with DNA to form structures called nucleosomes. If both strands of a DNA double helix break in the same place, this leads to a molecule called ubiquitin being attached to a histone called H2A within a nucleosome to mark the position of the damage. This promotes DNA repair by attracting another protein called RNF169 to bind to the nucleosome.

The precise location of the ubiquitin molecule on histone H2A is important because ubiquitin molecules act as signals for a variety of different processes when attached to specific positions on histones and other proteins. For example, ubiquitin molecules attached to some sites on histones can alter how the cell uses the genetic information contained within the nucleosome. However, it is not clear how the number and precise locations of ubiquitins on histones can produce such different signals.

Kitevski-LeBlanc, Fradet-Turcotte et al. investigated why RNF169 is only attracted to nucleosomes when ubiquitin is attached to a particular site on histone H2A following damage to DNA. The experiments reveal that two regions of RNF169 known as arginine motifs play an important role in controlling when the protein binds to nucleosomes. These arginine motifs – which are next to the region of the protein that binds to ubiquitin – identify the position of the ubiquitin on H2A by making contact with an "acidic" patch on the surface of the nucleosome.

These findings show that the combination of RNF169 binding to both the ubiquitin on H2A and an acidic patch on the nucleosome ensure that this protein only promotes DNA repair when and where it is needed. This acidic patch is involved in regulating the binding of various other proteins to nucleosomes. Understanding how cells interpret the signals produced by ubiquitin binding to proteins will help us to understand how disrupting these signals can contribute to cancer and other diseases.

formation of an isopeptide bond between the terminal carboxyl group of ubiquitin and an acceptor protein. Conjugated ubiquitin can be targeted for additional ubiquitylation cycles, producing polyubiquitin chains of various lengths and topologies (*Komander and Rape, 2012*). Translating the resulting ubiquitin signal into a specific cellular response typically involves interactions between ubiquitin and proteins containing ubiquitin-binding domains (UBDs) (*Dikic et al., 2009*). UBDs, which are small (20–150 amino acids) structurally diverse protein modules that interact with ubiquitin in a non-covalent manner, have been identified in more than 150 cellular proteins (*Hicke et al., 2005*). The specificity of UBD-ubiquitin interactions is achieved by a variety of mechanisms that can involve distinct affinities for certain linkages and lengths of ubiquitin chains, avidity through combination of UBDs, and contributions by UBD-independent sequences within a ubiquitin binding protein (*Dikic et al., 2009*). This latter mechanism is especially prevalent during the response to DSBs where some proteins harbor one such 'specificity' sequence adjacent to the UBD, termed an LR-motif or LRM (*Panier et al., 2012*).

The DSB response is a useful model to study ubiquitin-based signaling. In this process, DSBs elicit a complex chromatin modification cascade, illustrated in *Figure 1A*, which ultimately controls DNA repair (*Panier and Durocher, 2013*). The cascade begins with the phosphorylation of the C-terminal tail of histone H2AX, by the phosphatidylinositol 3-kinase-related kinase ATM and related kinases. Phosphorylated H2AX (known as γ-H2AX) is 'read' by the BRCT domains of the MDC1, which is itself a target of ATM. The ubiquitin ligase RNF8 then binds ATM-phosphorylated MDC1 where, together with the E2-conjugating enzyme UBC13, it catalyzes K63-linked polyubiquitin chains on linker histone H1. These chains on H1 are recognized by the ubiquitin-dependent recruitment module 1 (UDM1) of RNF168 (*Figure 1B*), a second E3 ligase that plays a key function in DSB repair (*Thorslund et al.,*

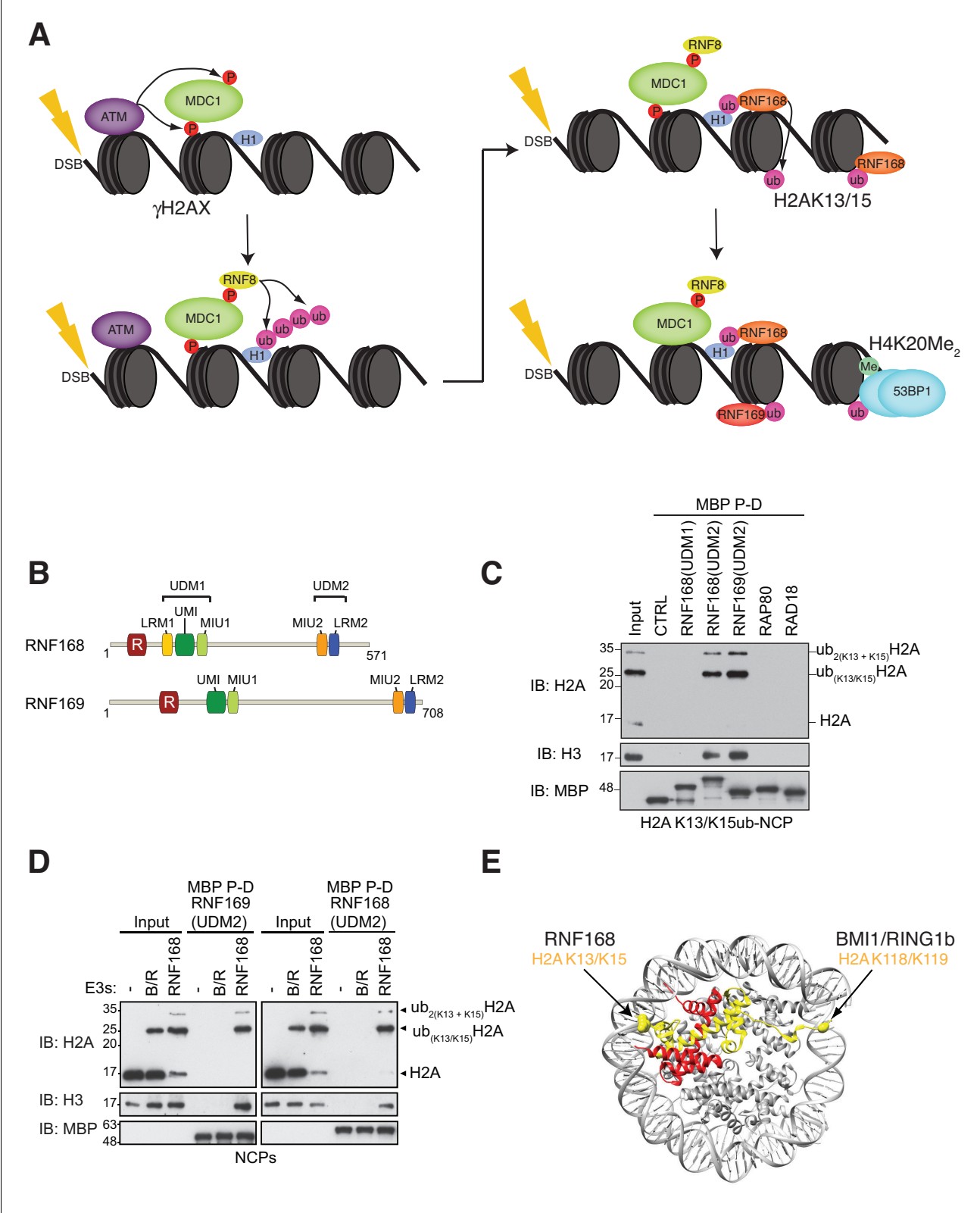

**Figure 1.** RNF168 and RNF169 bind RNF168-ubiquitylated NCPs. (**A**) Schematic of RNF8-mediated DNA DSB repair pathway. ATM: Ataxia telangiectasia mutated, MDC1: mediator of DNA damage checkpoint 1, BRCT: breast cancer 1 C-terminal, H1: linker histone H1, RNF: ring finger proteins, 53BP1: p53 binding protein 1, Ub: ubiquitin, P: phosphate group, Me: methyl group. (**B**) Domain architecture of RNF168(1-571) and RNF169(1-708). Domains and motifs are indicated. R: RING domain, MIU: motif interacting with ubiquitin, UIM: ubiquitin-interacting motif, UMI: UIM-, MIU-related

*Figure 1 continued on next page*

*Figure 1 continued*

ubiquitin binding motif, LRM: LR motif. (C) MBP pull-down assays of RNF168-ubiquitylated nucleosome core particles (H2AK13/K15ub-NCP) with the indicated MBP fusion proteins (RNF168-UDM1(110–201), RNF168-UDM2(374–571), RNF169-UDM2(662–708), RAP80(60-124) and RAD18(201-240)). Input: 5% of the amount of ubiquitylated NCPs used in the pull-down. The migration of molecular mass markers (kDa) is indicated on the left. (D) Pull-down assays of NCPs ubiquitylated with the indicated E3s by either MBP–RNF169(UDM2) (left) or MBP–RNF168(UDM2) (right). A reaction without E3 (-) acts as a negative control. B/R: BMI1/RING1b. (E) Structure of the nucleosome (PDB: 2PYO [*Clapier et al., 2008b*]). One copy of H2A and H2B is labeled in yellow and in red, respectively. Lysines that are ubiquitylated by RNF168 (H2A K13/K15) and BMI1/RING1B (H2A K118/K119) are indicated in space filling representation.

The following figure supplement is available for figure 1:

**Figure supplement 1.** RNF169(UDM2) binds with higher affinity than RNF168 and does not discriminate between H2AK13 and K15 ubiquitylation.

*2015*). The RNF168-UDM1 module is composed of two ubiquitin binding domains, MIU1 (motif interacting with ubiquitin) and UMI (MIU- and UIM-related ubiquitin binding motif) along with the LRM1 specificity module. RNF168 catalyzes mono-ubiquitylation of H2A/H2AX on Lys13 or Lys15 yielding H2AK13ub and H2AK15ub, respectively (*Gatti et al., 2012*; *Mattiroli et al., 2012*). The action of RNF168 on chromatin results in the subsequent recruitment of additional DSB repair factors including 53BP1, RAP80, RAD18 and the RNF168 paralog RNF169. RNF168 also promotes its own recruitment to DSB sites via a second UDM module, UDM2, consisting of the MIU2 UBD and the LRM2 specificity motif (*Figure 1B*). In contrast, RNF169 only contains a functional UDM2 module (*Figure 1B*), explaining the strict dependence on RNF168 for its recruitment to DSB sites (*Panier et al., 2012*; *Chen et al., 2012*; *Poulsen et al., 2012*).

While 53BP1, RNF168, RNF169, RAP80 and RAD18 accumulate on the chromatin surrounding DSB sites in an RNF168- and ubiquitin-dependent manner, how these factors recognize the products of RNF168 action are only beginning to emerge. 53BP1, while lacking a recognizable UBD, specifically interacts with H2AK15ub, but not with H2AK13ub, in the context of nucleosomes also containing H4 Lys20 dimethylation (*Fradet-Turcotte et al., 2013*). The structural basis for the site-specific recognition of H2AK15ub/H4K20me2 marks by 53BP1 was recently elucidated by electron cryomicroscopy (cryo-EM) (*Wilson et al., 2016*). The resulting structure showed that a short 53BP1 peptide segment, the ubiquitin-dependent recruitment (UDR) motif, interacts with the nucleosome core particle (NCP) surface, sandwiched by a ubiquitin molecule (*Wilson et al., 2016*). Like 53BP1, both RNF168 and RNF169 interact with ubiquitylated H2A (*Panier et al., 2012*), but it is not known if this interaction is selective for H2AK13/K15ub.

Here, we first establish that RNF168 and RNF169 are site-specific readers of H2AK13/K15 ubiquitylation. Then, we make use of an integrative approach to elucidate the molecular basis for the specific recruitment of RNF168/169 to DSBs by generating a structural model of the RNF169(UDM2)-ubiquitylated nucleosome complex. Central to these efforts have been methyl-TROSY NMR, which enables the study of protein complexes with aggregate molecular masses as large as 1 MDa (*Sprangers and Kay, 2007*; *Rosenzweig and Kay, 2014*; *Gelis et al., 2007*), and replica averaged molecular dynamics simulations (*Cavalli et al., 2013a*) using restraints derived from NMR and site-directed mutagenesis experiments. Our model establishes that the previously identified bipartite interaction module of RNF169 (*Panier et al., 2012*) is composed of a classic MIU helix, that docks onto the canonical hydrophobic surface of ubiquitin, while the LRM2 region responsible for specificity is highly disordered. The LRM2 motif binds to the acidic patch region on the nucleosome surface using two basic regions with key arginine residues. We then use cryo-EM to validate this model. These results demonstrate the synergistic behavior of MIU-ubiquitin and LRM-NCP acidic patch interactions in facilitating recruitment of histone ubiquitylation readers RNF168/RNF169 to regions of DSBs and in providing the specificity of the interaction.

## Results

### The RNF169 UDM2 module interacts with H2AK13/K15ub NCPs

To identify additional proteins that specifically recognize H2AK13/K15ub in the context of nucleosomes, we prepared MBP fusion proteins with the RNF168 UDM1 and UDM2 modules, the RNF169 UDM2 module and regions of RAD18 and RAP80 that are necessary and sufficient for their ubiquitin-dependent recruitment to DSB sites (*Panier et al., 2012*). We observed that the RNF168 and RNF169 UDM2 modules were the only proteins, among those tested, that bound to NCPs catalytically ubiquitylated with the RNF168(1-113)-UBCH5a complex in MBP pull-down assays (*Figure 1C*). These data suggest that the RNF168 and RNF169 UDM2 modules might represent selective readers of H2A Lys-13/Lys-15 ubiquitylation.

H2A is the most abundant ubiquitylated protein in the nucleus with 5–15% of H2A conjugated to ubiquitin at the K119 residue in vertebrate cells (*Cao and Yan, 2012*). As the BMI1-RING1b complex can catalyze this mark (*Wang et al., 2004*), we tested whether the RNF168 or RNF169 UDM2 modules interact with H2AK119ub-containing NCPs prepared using the BMI1-RING1b enzyme complex. While both RNF169(UDM2) and RNF168(UDM2) were able to pull-down H2A K13/K15ub-NCPs, they were unable to interact with BMI1/RING1b-ubiquitylated NCPs (*Figure 1D*). Although both NCP substrates provide accessible ubiquitin molecules, since they are attached to unstructured histone tails, the ubiquitin moieties reside on opposite ends of the NCP surface (*Figure 1E*). Therefore, these results imply that the UDM2 module makes specific contacts with the N-terminal region of H2A or with the surrounding NCP surface, rather than non-specific interactions with DNA, to establish a strong and selective association with ubiquitylated NCPs.

### Solution NMR of RNF169(UDM2) and its binding to ubiquitin and to ubiquitylated NCPs

Having identified an interaction module required for binding of H2AK13/K15ub-NCPs, involving both MIU and LRM domains, we then used solution NMR spectroscopy to establish the secondary structures of these elements in the unbound state. The primary sequences of RNF169(UDM2) and RNF168(UDM2), encompassing the MIU2 and LRM2 domains, are shown in *Figure 2A*. Residues 665–682 of RNF169 and 442–459 of RNF168 comprise the conserved primary sequence specific to MIUs, while the 13 residue long LRM2 regions contain the critical upstream arginine (R689 in RNF169 and R466 in RNF168) and the LR pair (L699/R700 in RNF169 and L476/R477 in RNF168) required for the stable interaction with NCPs ubiquitylated by RNF168 (*Panier et al., 2012*). Despite the fact that RNF168 and RNF169 have high sequence similarity within their analogous MIU2-LRM2 modules, we observed a more robust interaction between RNF169(UDM2) and H2AK13/K15ub-NCPs in MBP pull-down experiments (*Figure 1—figure supplement 1A*). Consequently, we focused our NMR experiments and subsequent structural models on the RNF169(UDM2)-H2AK13ub-NCP interaction since we ascertained that the UDM2 module, unlike 53BP1, can bind to either H2AK13ub or H2AK15ub (*Figure 1—figure supplement 1B*) (*Fradet-Turcotte et al., 2013*). Assignment of RNF169(UDM2) backbone resonances was carried out using standard triple-resonance solution NMR experiments (*Sattler et al., 1999*) at 35°C, with the assigned $^1$H-$^{15}$N HSQC spectrum shown in *Figure 2B*. The secondary structural propensity program SSP was used to determine the type and probability of secondary structural elements based on measured $^1$Hα, $^{13}$Cα, $^{13}$Cβ, and $^{13}$CO chemical shifts (*Marsh et al., 2006*). The results, shown in *Figure 2C*, indicate a relatively high propensity for helical secondary structure over much of the MIU2 region (665-682) and some propensity for helix into the region linking the MIU2 and LRM2 domains (683-685). The remaining linker region, LRM2, and residues C-terminal to the LRM2 appear to lack any secondary structure, aside from a modest increase in helical propensity peaking at residues Y697 and L698, with an average 28% helical probability. These two residues are dynamic with Chemical Shift Index (CSI) based order parameters (*Berjanskii and Wishart, 2005*) of 0.56 and 0.46, respectively, where the order parameter is a metric used to quantify the amplitude of backbone motion, ranging from 0 (isotropic motion) to 1 (rigid) (*Lipari and Szabo, 1982*). In contrast, an average value for CSI-based order parameters of 0.80 ± 0.04 is obtained for the MIU2 sequence, indicating that it is significantly more rigid. On the basis of MIU2 sequence conservation and the formation of helical structure in the MIU2 region, it is likely that the MIU2 of RNF169(UDM2) interacts with ubiquitin via a classic MIU-ub interaction,

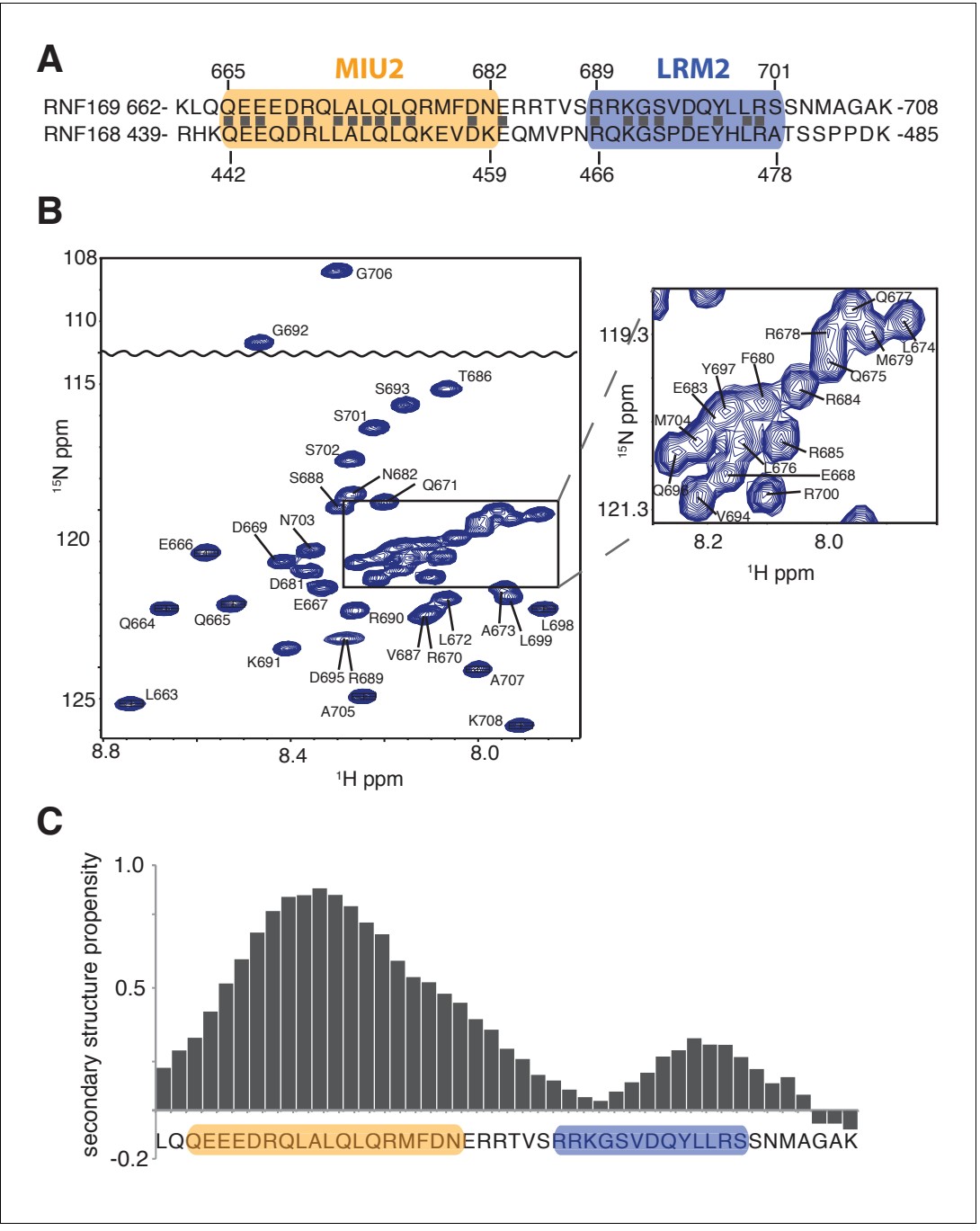

**Figure 2.** Solution NMR analysis of RNF169(UDM2) reveals a flexible LRM2. (**A**) Primary sequence of MIU2 (orange) and LRM2 (blue) modules of both RNF169 and RNF168. Conserved residues are indicated with vertical bars. (**B**) Assigned $^1$H-$^{15}$N HSQC spectrum of $^{15}$N,$^{13}$C-labeled RNF169(UDM2). Data collected at 11.7 T, 35°C. (**C**) Output from SSP program (*Marsh et al., 2006*) using RNF169(UDM2) backbone chemical shifts as input. Corresponding primary sequence of RNF169(662-708) displayed along horizontal axis, with MIU2 (orange) and LRM2 (blue) regions highlighted. Positive and negative SSP values indicate α-helical and β-strand secondary structure propensities, respectively.

The following figure supplement is available for figure 2:

**Figure supplement 1.** Modified RNF169(UDM2) construct exhibits higher thermo stability.

whereby the helical arrangement aligns critical hydrophobic residues on one face of the helix so as to form contacts with the canonical hydrophobic patch of ubiquitin (*Penengo et al., 2006*).

In order to improve the thermal stability of the MIU2 helix, we engineered an RNF169 protein containing the UDM2 sequence as before but where an aspartic acid, acting as a N-terminal helix cap, was introduced, followed by three alanine residues prior to K662, *Figure 2A* (*Figure 2—figure supplement 1A*). This peptide was used in all NMR studies with NCPs, since it tolerates temperatures of 45°C, where spectral quality is improved. It is noteworthy that the helical content of the thermostable variant at 45°C is near that of the wild-type peptide at 35°C, as shown by circular dichroism spectroscopy (*Figure 2—figure supplement 1B*).

To experimentally establish the molecular basis for the interaction between the MIU2 of RNF169 and ubiquitin, we performed NMR-based titrations of isotope-labeled ubiquitin, both free and chemically-attached to reconstituted nucleosomes. Chemical ubiquitylation of H2A at position K13 was achieved by converting G76 of ubiquitin and K13 in H2A to cysteine residues followed by conjugation via a sidechain-sidechain disulfide linkage (*Figure 3—figure supplement 1A*). The ubiquitylated histone was then assembled into octamers and wrapped with DNA in the absence of any reducing agent to produce H2AK13Cub-NCPs, as previously described (*Dyer et al., 2004*). The significant mass difference between free ubiquitin and ubiquitin conjugated to NCPs enabled the use of NMR-based $^{13}$C-edited diffusion experiments (*Choy et al., 2002*), focusing on ubiquitin labeled with $^{13}CH_3$ methyl groups (see below), to confirm the conjugation product and monitor sample stability (*Figure 3—figure supplement 1B*). The diffusion constant so obtained, D = 3.4 ± 0.2 × $10^{-7}$ cm²/s at 25°C, is consistent with a particle of molecular mass of approximately 250 kDa and is slightly smaller than for the 1/4 proteasome comprising a heptameric ring of α-subunits (180 kDa) (*Religa et al., 2010*) where D = 4.2 ± 0.1 x $10^{-7}$ cm²/s has been measured under identical conditions. That the C76ub-C13H2A sidechain disulfide linkage supports RNF169 binding was demonstrated by showing that MBP-RNF169(UDM2) pulled down H2AK13Cub-NCPs as efficiently as catalytically prepared H2AK13ub-NCPs containing the isopeptide linkage (*Figure 3—figure supplement 1C*).

Having established both the suitability and the stability of our H2AK13Cub-NCPs, we carried out NMR titration experiments with RNF169(UDM2) (unlabeled RNF169 in the case of free ubiquitin and perdeuterated for NCPs) using $^{13}$C-labeled free ubiquitin or highly deuterated, Ile-δ1-$^{13}CH_3$, Leu, Val-$^{13}CH_3$,$^{12}CD_3$-ubiquitin (referred to as ILV-methyl labeled in what follows) attached to H2AK13C-NCPs (perdeuterated histone NCPs). *Figure 3A* shows a series of overlaid $^1$H-$^{13}$C HMQC spectra of ILV-methyl labeled ubiquitin in H2AK13Cub-NCPs as a function of increasing amounts of RNF169 (UDM2). Chemical shift perturbations (CSPs) in both free (*Figure 3—figure supplement 2A*) and NCP-bound ubiquitin were of the same direction and magnitude, indicating very similar interactions for RNF169(UDM2) with both free and NCP-bound ubiquitin. Peaks exhibiting the largest CSPs are derived from residues that contribute to the canonical ubiquitin hydrophobic patch, including I44, L8 and V70 (*Dikic et al., 2009*). Indeed, NCPs conjugated to ubiquitin harboring an I44A point mutation failed to interact with the UDM2 module of both RNF168 and RNF169 in MBP pull-downs (*Figure 3—figure supplement 2B*). CSP-derived binding curves were fit to extract equilibrium dissociation constants for RNF169(UDM2) with free ubiquitin and H2AK13Cub-NCPs of 956 ± 340 μM, 45°C, and 24 ± 7 μM, 45°C, respectively (*Figure 3B*). It is noteworthy that only residues with relatively small CSPs, corresponding to fast exchange on the NMR chemical shift timescale, were included in the analysis since in this limit chemical shifts are directly related to dissociation constants ($K_D$) and are not influenced by kinetics, allowing reliable estimates of affinity from peak positions. The low affinity interaction between RNF169(UDM2) and free ubiquitin is characteristic of ubiquitin binding domains, having dissociation constants generally larger than 100 μM (*Hurley et al., 2006*). The presence of the NCP increases the affinity by approximately 40-fold, presumably through specific contacts between the LRM2 and the NCP surface near K13 of H2A. The use of two cooperative, relatively low affinity interactions to impart specificity is a common theme of ubiquitin signaling (*Chen et al., 2009*).

Notably, for I44, that has the largest CSP upon binding (~126 Hz), exchange between RNF169 (UDM2) free and bound NCP states is in the intermediate regime on the chemical shift timescale, facilitating the extraction of kinetic parameters via lineshape analysis (*Tugarinov and Kay, 2003*). *Figure 3C* shows a comparison of experimental lineshapes (black) extracted from the $^{13}$C dimension of $^1$H-$^{13}$C HMQC spectra for I44 (δ1 methyl) as a function of increasing amounts of added RNF169

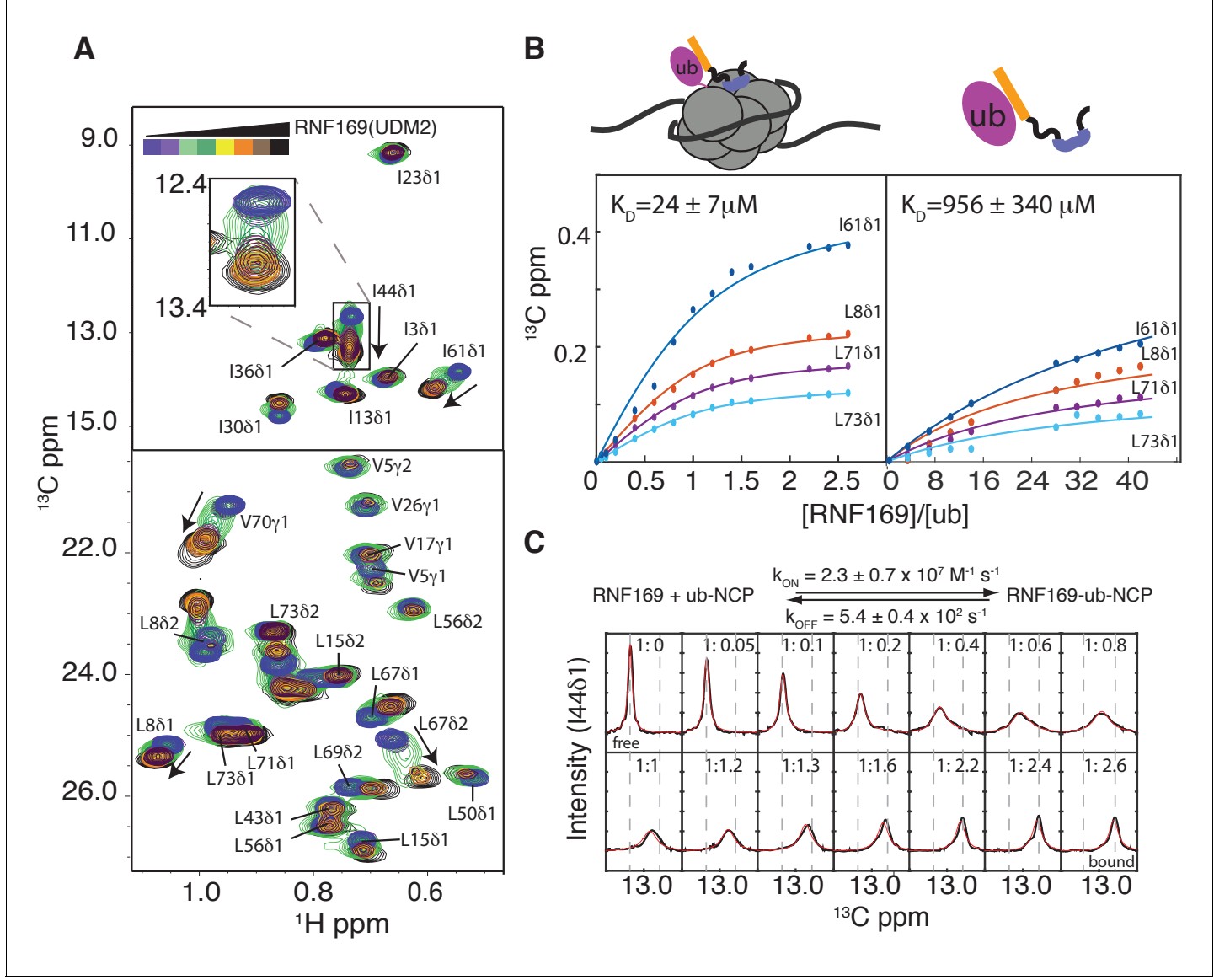

**Figure 3.** Thermodynamics and kinetics of the RNF169(UDM2)-H2AK13C ub-NCP interaction. (A) Selected regions of $^1$H-$^{13}$C HMQC spectra of ILV-methyl labeled ubiquitin in H2AK13Cub-NCPs with increasing amounts of unlabeled RNF169(UDM2). Arrows indicate direction of peak movement. Data collected at 14.1 T, 45°C. (B) Chemical shift derived binding curves for selected residues in ILV-methyl labeled ubiquitin in H2AK13Cub-NCPs (left panel) and free ILV-methyl labeled ubiquitin (right panel) upon addition of unlabeled RNF169(UDM2) (circles), with best fits (solid lines) shown. Ratio of RNF169(UDM2) to ubiquitin indicated on horizontal axis. (C) Fitted line shapes for I44δ1 (extracted from $^1$H-$^{13}$C correlation spectra by taking traces along the $^{13}$C-dimension). Experimental data in black, with simulated line shapes in red. Ratio of ubiquitin to RNF169(UDM2) indicated in each panel and vertical grey dashed lines denote the resonance positions of I44δ1 in the absence (free) and presence (bound) of saturating amounts of RNF169 (UDM2). The extracted kinetic parameters for the RNF169, ub-NCP binding reaction are shown above the traces.

The following figure supplements are available for figure 3:

**Figure supplement 1.** Preparation and validation of disulfide linked H2AK13Cub-NCPs.

**Figure supplement 2.** RNF169(UDM2) binds ubiquitin through its canonical hydorphobic face in both the free and NCP-bound context.

**Figure supplement 3.** Selected regions of CT-$^1$H-$^{13}$C HSQC spectra of $^{15}$N,$^{13}$C wild-type RNF169(UDM2) with increasing amounts of unlabeled ubiquitin.

(UDM2), with fitted simulated lineshapes that include the effects of chemical exchange in red. Using the value of $K_D$ = 24 ± 7 µM estimated from the titration data, on and off-rates are fit to be 2.4 ± 0.7×10⁷ M⁻¹ s⁻¹ and 5.4 ± 0.4×10² s⁻¹ respectively. Notably, the RNF169(UDM2) on-rate is an order of magnitude faster than the diffusion limit (*Alsallaq and Zhou, 2007*; *Schlosshauer and Baker, 2004*), emphasizing the importance of electrostatic interactions in contributing to the association.

## Methyl-TROSY NMR reveals RNF169(UDM2) binding sites on the NCP

To investigate the regions of the NCP involved in the interaction with the LRM2, we monitored CSPs in ¹H-¹³C HMQC spectra upon addition of RNF169(UDM2). As discussed previously methyl-HMQC spectra are significantly enhanced in sensitivity and resolution by a methyl-TROSY effect that preserves signal (*Tugarinov et al., 2003*). This aspect is of particular importance in applications to high molecular weight proteins and protein complexes (*Sprangers and Kay, 2007*; *Rosenzweig and Kay, 2014*; *Gelis et al., 2007*), such as the NCP (*Kato et al., 2011a*). *Figure 4A* shows ¹H-¹³C HMQCs of ILV-methyl labeled H2A and H2B in the context of H2AK13Cub-NCPs, unbound or with

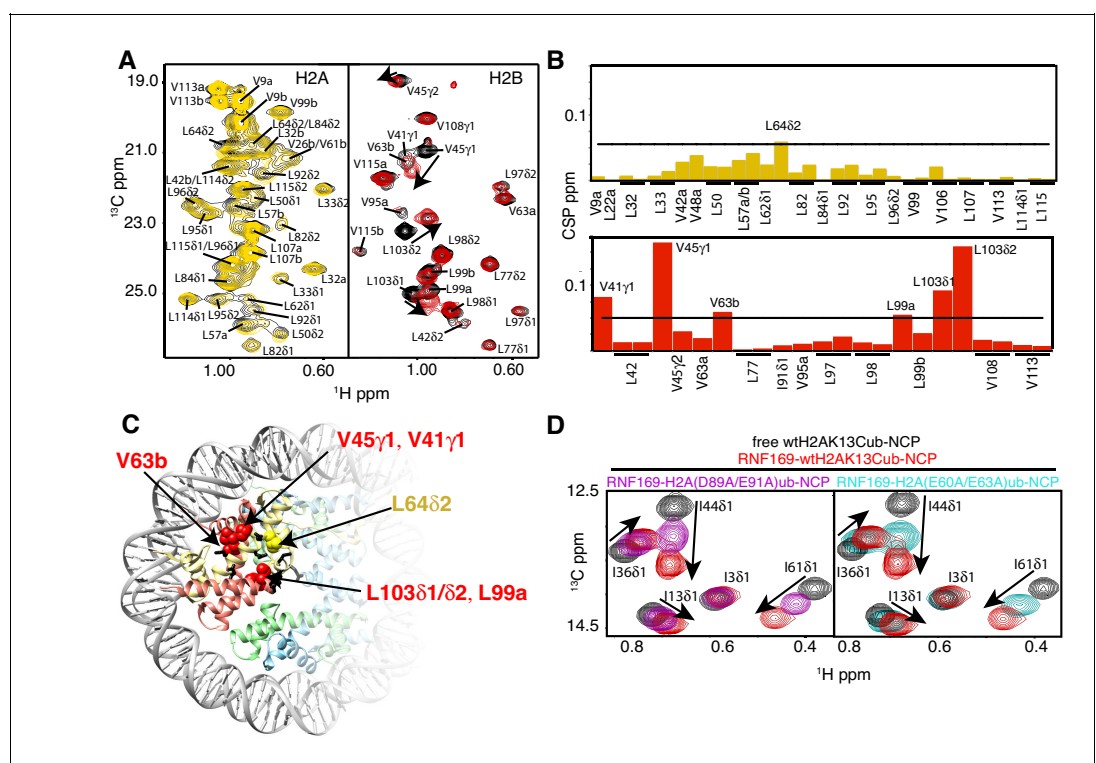

**Figure 4.** NMR and mutagenesis identify the nucleosome acidic patch as the binding interface for RNF169(UDM2). (**A**) Superimposed ¹H-¹³C HMQC spectra of ILV-methyl labeled H2A (left panel) and ILV-methyl labeled H2B (right panel) in the context of the H2AK13Cub-NCP without (black) and with (yellow, left; red, right) wild-type RNF169(UDM2), respectively. RNF169(UDM2) was added at 2.5-fold excess relative to ubiquitin. Arrows indicate peak movement. Data collected at 14.1 T, 45°C. (**B**) Chemical shift perturbations (CSPs) in ILV-methyl labeled H2A (yellow, top panel) and ILV methyl-labeled H2B (red, bottom panel) H2AK13Cub-NCPs. Residues with CSP values 1σ above the average are indicated (black line). CSPs were calculated as described in Materials and Methods. (**C**) Location of residues with significant CSPs in H2A (yellow) and H2B (red) shown in space filling representation and indicated with arrows on nucleosome crystal structure (2PYO) (*Clapier et al., 2008a*). Acidic patch residues are shown in stick representation and coloured black. H2A: light yellow, H2B: salmon, H3: light blue and H4: light green. (**D**) Selected isoleucine regions of ¹H-¹³C HMQC spectra of free (black) and 2.5-fold excess RNF169(UDM2) bound (red) ILV-methyl labeled ub H2AK13Cub-NCP. Spectra of acidic patch mutant NCPs, H2AK13C(D89A/E91A)ub-NCP (purple, left panel) and H2AK13C(E60A/E63A)ub-NCP (teal, right panel), with 5-fold excess RNF169(UDM2) to ubiquitin are overlaid and highlight the resulting binding deficiency. All data are recorded at 14.1 T, 45°C.

The following figure supplement is available for figure 4:

**Figure supplement 1.** Overlay of ¹H-¹³C HMQC spectra of ILV-methyl labeled H3 H2AK13Cub-NCPs with (red) and without (black) RNF169(UDM2).

saturating amounts of RNF169(UDM2). Corresponding weighted CSPs are tabulated in *Figure 4B* (see Materials and methods). Upon binding RNF169(UDM2) L64δ2 of H2A and V41γ1, V45γ1, V63b, L99a and L103δ1/δ2 of H2B showed significant CSPs. Note that in the absence of stereospecific assignments isopropyl methyl groups are referred to as 'a' or 'b' to denote upfield and downfield resonating moieties. Interestingly, the methyl-bearing residues mentioned above surround a highly negative surface of the H2A/H2B dimer, known as the acidic patch. The acidic patch is comprised of eight residues in total; six in H2A (E55(E56), E60(E61), E63(E64), D89(D90), E90(E91) and E91(E92) in *Drosophila melanogaster* (*Homo sapiens*)) and two in H2B (E102 and E110), creating a contoured and highly charged groove (*Kalashnikova et al., 2013*). The acidic patch is a well-characterized interaction surface for a multitude of NCP binding proteins with a broad range of functions (*Kato et al., 2011a*; *Barbera et al., 2006*; *McGinty et al., 2014*; *Makde et al., 2010*; *Armache et al., 2011*; *Morgan et al., 2016*). Interestingly, no significant CSPs were observed in H3 (*Figure 4—figure supplement 1*), consistent with localized interactions involving the H2A/H2B dimer face of the NCP (*Figure 4C*).

While the methyl-TROSY NMR results implicate the acidic patch surface as the main contact point between the NCP and LRM2, we made use of a combined site-directed mutagenesis – NMR approach to directly monitor the effect of acidic patch residues. Specifically, we prepared H2AK13Cub-NCPs with pairwise mutations in H2A including D89A/E91A and E60A/E63A, effectively neutralizing two regions of the acidic patch surface. The first mutant, H2A D89A/E91A, alters a region of the acidic patch that is responsible for making several contacts with the canonical residue common to all known acidic patch-binding proteins (*McGinty and Tan, 2015*). The second mutant, H2A E60A/E63A, was designed to neutralize the region of the acidic patch closer to the DNA, where additional contacts have been identified in NCP-binding proteins (*Kalashnikova et al., 2013*). The large chemical shift changes of isoleucine residues in ubiquitin upon binding RNF169(UDM2) provide spectral signatures of free and RNF169(UDM2)-bound H2AK13Cub-NCPs, enabling a straightforward comparison of the binding capacity of the H2A mutants described above with wild-type H2A containing NCPs. The isoleucine region of ubiquitin in H2AK13C(D89A/E91A)ub- and H2AK13C(E60A/E63A)ub-NCPs, after addition of equivalent amounts of RNF169(UDM2), are shown in *Figure 4D*, overlaid with free (black) and RNF169(UDM2)-bound wild-type H2AK13Cub-NCP (red) spectra. For both H2AK13C(D89A/E91A)ub- and H2AK13C(E60A/E63A)ub-NCPs the binding capacity for RNF169(UDM2) is reduced considerably, with $K_D$ values of approximately 280 ± 100 μM and 230 ± 100 μM relative to 25 μM for binding to wild-type NCPs. The compromised binding displayed by the H2A mutant-NCPs provides strong validation of the methyl-TROSY CSP data and indicates that at least one member of each pairwise H2A mutant plays an important role in the interaction. Moreover, the mutagenesis data extend the range of NCP residues that can be investigated, since the NMR analysis is limited to only methyl-containing amino acids.

## Identification of key residues within the LRM2

Previous work has identified several highly conserved residues within the LRM2 of both RNF168 and RNF169 that are vital to their accumulation at DSBs (*Panier et al., 2012*). Individual substitutions of R689, Y697, and pairwise substitution of L699/R700 to alanine were found to abrogate recruitment of GST-RNF169(UDM2) to IR-induced foci (*Panier et al., 2012*). In agreement with these findings, R689A, Y697A, L699A and R700A were all found to reduce the capacity of RNF169 to inhibit 53BP1 focus formation after irradiation with a 2 Gy dose of X-rays (*Figure 5—figure supplement 1*).

While methyl-TROSY NMR was instrumental in identifying the acidic patch of the NCP surface as an important component in the interaction, a similar CSP-based analysis of the LRM2 was hampered by the presence of only three methyl-bearing residues in this motif (V694, L698 and L699) and by the complete resonance overlap of L698 and L699 (*Figure 5—figure supplement 2*). To confirm the results of the biological assay involving 53BP1 that established key LRM2 residues, we used a mutagenesis and methyl-TROSY NMR approach, similar to that described above. As before, we made use of the free and RNF169(UDM2)-bound signature spectra of isoleucine residues in ubiquitin to qualitatively evaluate the ability of single residue substitutions in RNF169(UDM2) to form a complex with wild-type H2AK13Cub-NCPs (*Figure 5*). As expected, R689, L699 and R700 substitutions reduce the ability of RNF169 to form a complex with H2AK13Cub-NCPs, while Y697A was also observed to impede binding to a significant degree, and S701A exhibited the smallest decrease in binding capacity.

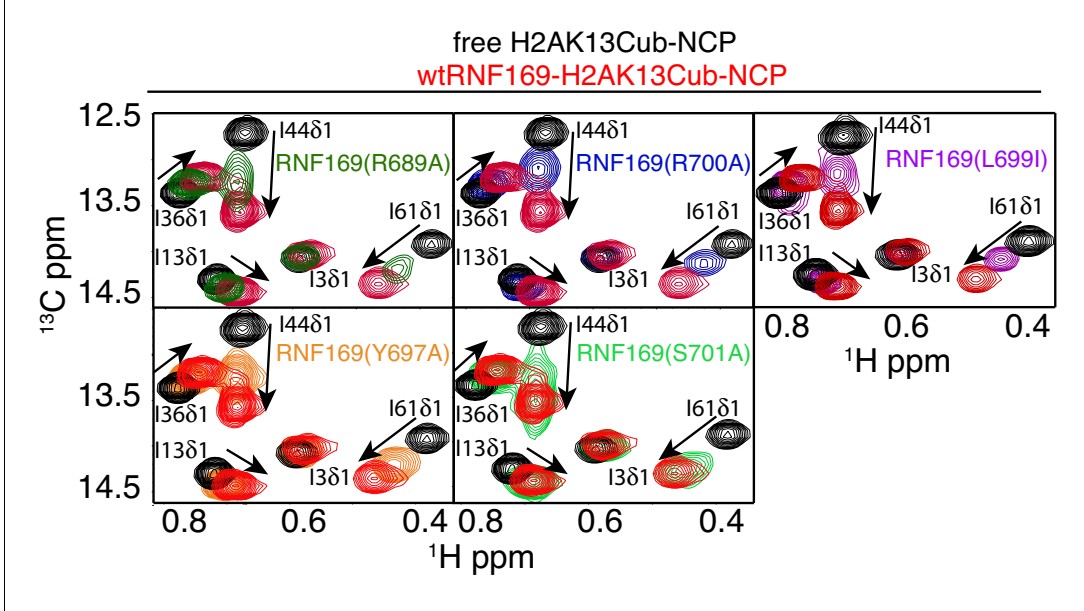

**Figure 5.** R689 and L699/R700 are critical to the formation of the complex. Isoleucine region of $^1$H-$^{13}$C HMQC spectra of ILV-methyl labeled ub H2AK13C-ubNCP without (black) and with (red) wild-type RNF169(UDM2) and the indicted RNF169(UDM2) LRM2 mutants. The ratio of wild-type or mutant RNF169(UDM2) to ubiquitin was 2.5:1 in all cases.

The following figure supplements are available for figure 5:

**Figure supplement 1.** Alanine scanning of LRM2 reveals criticial residues in RNF169(UDM2)-ubNCP interaction.

**Figure supplement 2.** Methyl resonances of LRM2 region are overlapped in $^1$H-$^{13}$C CT-HSQC of $^{15}$N,$^{13}$C wild-type RNF169(UDM2).

**Figure supplement 3.** NMR reveals a similar acidic patch binding mode for R689 and R700 of RNF169(LRM2) and LANA(1-23).

As mentioned above, the acidic patch is a common interaction surface for nucleosome binding proteins that typically involves key contacts between an arginine sidechain and the nucleosome surface (*Kato et al., 2011a*). The Kaposi's sarcoma herpes virus (KHSV) latency associated antigen (LANA) is an acidic patch binding protein with a critical LR-like motif (*Figure 5—figure supplement 3A*). MBP pull-down assays established that a LANA-derived peptide competes with RNF169 for H2AK13/K15ub-NCP binding with an estimated IC50 of 125–500 µM (*Figure 1—figure supplement 1A*) (*Barbera et al., 2006*). However, it was not clear *a priori* whether one or both of the basic stretches of residues in RNF169(UDM2) (R689-R690-K691 or L699-R700-S701) was important for binding and whether one mimicked the critical L8-R9-S10 sequence in LANA. A comparison of CSPs in ILV-labeled H2B upon binding RNF169(UDM2), LANA(1-23) and two RNF169(UDM2) triple mutants, (RNF169(R689A/R690A/K691A) and RNF169(L699A/R700A/S701A)), revealed similar chemical shift perturbation patterns between LANA and both RNF169 triple mutants (*Figure 5—figure supplement 3B*). While these results were unable to definitively identify the anchoring arginine in RNF169(UDM2), the fact that both triple mutants bound H2AK13Cub-NCPs reinforces the importance of both the RRK and LRS regions of RNF169(UDM2).

## Computational modeling reveals structural details of the RNF169 (UDM2) - H2AK13Cub-NCP complex

Our experimental data indicate the importance of three interactions for the selective recognition of H2AK13/K15ub-NCPs by RNF169(UDM2). The first, encompassing the MIU2 residues 665–682, is of low affinity, like for many MIU-ub complexes involving the canonical hydrophobic surface of ubiquitin (*Hurley et al., 2006*). This weak interaction is strengthened through two additional contacts

between the nucleosome acidic patch and the RRK and LRS regions of RNF169(UDM2) that provides selectivity for NCPs ubiquitylated on K13 or K15 of H2A.

As a first step to obtain the structure of the complex, we docked the MIU2 region of RNF169, corresponding to residues 662–682, onto ubiquitin (PDB ID 1UBQ) using experimental CSPs and a number of inter-molecular nuclear Overhauser effects (NOEs) as restraints (see Materials and Methods) with the HADDOCK molecular modeling program (*Dominguez et al., 2003*). Methyl chemical shift changes in the MIU2 region upon binding ubiquitin are localized to L672δ1/δ2, A673, and L676δ1/δ2, highly conserved MIU residues that are directly involved in the interaction (*Figure 6A*) (*Panier et al., 2012*; *Penengo et al., 2006*). Key NOEs connect A673 of the MIU2 to L8 and I44 of Ub, *Figure 6B*. The resulting HADDOCK model is shown in *Figure 6C* aligned with the X-ray structure of the Rabex MIU-ubiquitin complex (*Penengo et al., 2006*). Aside from a slight tilt in helix orientation (compare blue Rabex and gold RNF169 helices) the RNF169(665-682) MIU2-ub model is very similar to that previously published for the Rabex MIU-ub complex.

Having elucidated the structure of the MIU2-ub component of the complex, we next used cryo-EM in an attempt to obtain structural models of the ub-NCP particles in both the free and RNF169-bound states. Recent advances in single particle cryo-EM instrumentation and data processing have facilitated the calculation of high-resolution 3D-maps of biomolecules as small as 60 kDa (*Bai et al., 2015*; *Smith and Rubinstein, 2014*; *Khoshouei et al., 2016*). *Figure 7A–B* show cryo-EM density maps of free and RNF169-bound NCPs, determined at 8.1 and 6.6 Å resolution, respectively (*Figure 7—figure supplement 1*). In both cases the rigid NCP forms the symmetrical discoid shape expected from NCP x-ray structures (*Clapier et al., 2008a*). A striking difference between the two maps is the density corresponding to ubiquitin, which at the threshold level used, is only visible in the RNF169-bound state. In the absence of RNF169(UDM2) ubiquitin is highly dynamic, occupying a variety of positions with respect to the NCP due to its conjugation to the flexible N-terminal tail of H2A. The density in this portion of the map is thus low in comparison to the NCP core itself. These results are illustrated in *Figure 7C* using equivalent lateral slices through free and RNF169-bound NCP cryo-EM maps, with the raw map density colored according to local resolution estimates (*Kucukelbir et al., 2014*). As expected, the NCP core in both maps exhibits the highest resolution, while the resolution of the ubiquitin region of the map is significantly lower in the absence of RNF169(UDM2). The restricted motion of ubiquitin within the complex can also be established by measuring NMR spin relaxation rates of ubiquitin methyl probes in H2AK13Cub NCPs in the presence and absence of RNF169. Here we have quantified intra-methyl $^1$H-$^1$H dipolar cross-correlated relaxation rates (*Sun et al., 2011*), focusing on ILV residues. In the macromolecular limit these rates are proportional to the product $S^2\tau_C$, where $S^2$ is the square of an order parameter describing the amplitude of motion for the 3-fold symmetry axis of the methyl group, and $\tau_c$ is the molecular tumbling time that in the present case provides a measure for how rigidly attached ubiquitin is to the NCP. Residue-specific $S^2\tau_c$ values are plotted in *Figure 7D* for ub-NCP (blue) and for RNF169 (UDM2) ub-NCP (pink) with an average 2-fold increase, reflecting a reduction in conformational space available to ubiquitin upon addition of RNF169(UDM2).

While the cryo-EM map defined aspects of the overall topology of the complex, higher resolution information is required to obtain an atomic level description of key interactions that provide specificity and, in particular, to resolve the role of the important arginine residues, R689 and R700. We used replica-averaged molecular dynamics simulations to develop a structural model consistent with our experimental NMR and mutagenesis data (*Cavalli et al., 2013a*; *Kukic et al., 2014a*, *2016*). In this method, experimental data are incorporated during the simulations as replica-averaged restraints whereby back-calculated parameters are compared with their experimentally measured values to evolve the system in a manner such that the agreement with the experimental restraints increases over time. This approach is described in detail in Materials and methods and illustrated schematically in *Figure 8—figure supplement 1*.

*Figure 8A* shows an overlay of ten members from the calculated ensemble of approximately 600 structures. Enlarged views of a pair of structures, focusing on the region of contact between the LRM2 (blue) and acidic patch residues (yellow and red), are highlighted in *Figure 8B*. Notably, in the great majority of structures the LRM2 backbone remains highly disordered within the complex and does not form regular secondary structure. In less than 5% of the conformers the LRM2 region contains either a small antiparallel $\beta$ sheet encompassing residues Y697 to M704, or a small 3-residue helix involving residues between Y697 and M704. Importantly, in all members of the ensemble R700

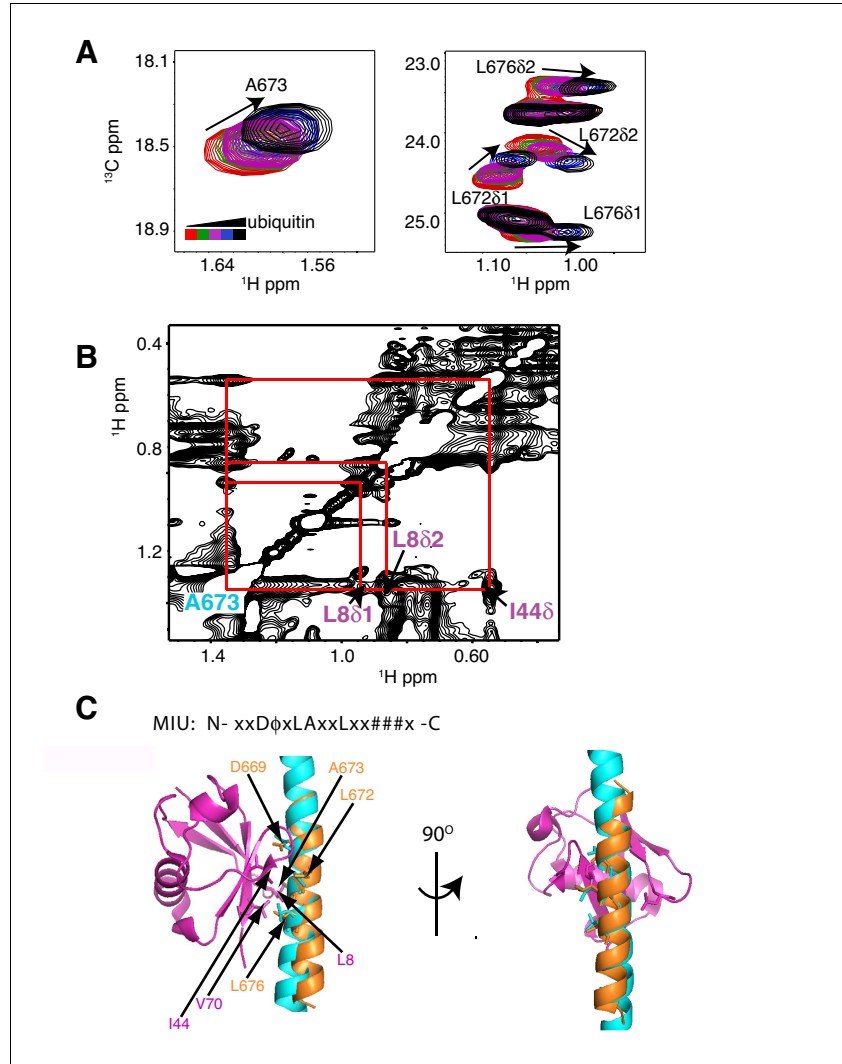

**Figure 6.** The RNF169 MIU2-ubiquitin interaction involves the canonical binding surface of ubiquitin and a central alanine in the MIU2. (**A**) Overlaid regions of $^1H$-$^{13}C$ CT-HSQC spectra of $^{13}C$-labeled RNF169(UDM2) upon addition of increasing amounts of free ubiquitin. Residues with significant chemical shift changes are labeled. Data recorded at 11.7 T, 35°C. (**B**) Selected region of $^1H$-$^1H$ NOESY spectrum of ILV-methyl labeled ubiquitin and $^{13}C$-labeled RNF169(UDM2) at 1:8 molar ratio (200 ms mixing time). Cross peaks between A673 of RNF169(UDM2) and I44δ1, L8δ1/δ2 of ubiquitin are indicated. Data recorded at 14.1 T, 20°C. (**C**) Signature MIU primary sequence; x: any amino acid type, φ: large hydrophobic and #: acidic. Structural model of RNF169(MIU2)-ubiquitin from HADDOCK docking calculations, aligned for comparison with crystal structure of the Rabex(MIU)-ubiquitin complex (2C7N (*Penengo et al., 2006*)). Signature MIU residues within RNF169(MIU2) and Rabex and hydrophobic patch residues are shown in stick representation. Ubiquitin: magenta, RNF169(MIU2): orange, Rabex (MIU): cyan.

is consistently located in the position where it makes contacts with one or more of the key acidic patch residues E60, D89 and E91. R689 also contacts the acidic patch in all structures through interactions involving at least one of E60 (H2A), E63 (H2A) or D48 (H2B). As shown in *Figure 8B*, the position of R700 within the ensemble is relatively fixed, while R689 alternates between positions that permit electrostatic contacts with E63 or D48. Although R700 and R689 were shown by mutagenesis to be almost equally important for formation of the RNF169(UMD2)-ubNCP complex (see above), our replica-averaged MD results indicate that R700, rather than R689, is the critical anchoring arginine. Mutagenesis and NMR studies also indicate a significant role for L699 and, to a lesser extent for Y697 in complex formation (*Figure 5A*). For a significant subset of the structural ensemble

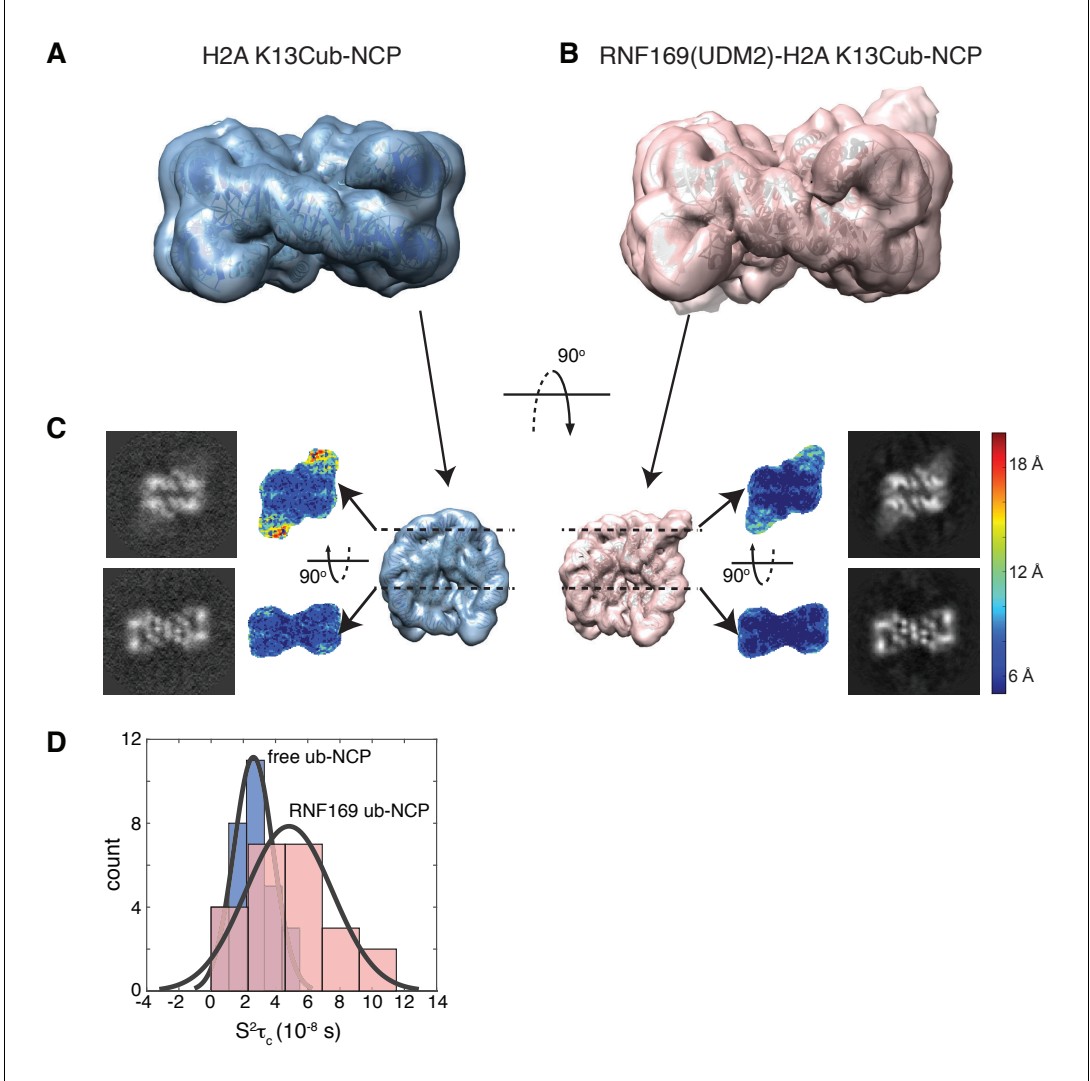

**Figure 7.** Ubiquitin is highly dynamic in the absence of RNF169(UDM2). (**A**) H2AK13Cub-NCP cryo-EM map at 8.1 Å resolution and (**B**) RNF169(UDM2) bound H2AK13Cub-NCP cryo-EM map at 6.6 Å resolution including the drosophila NCP crystal structure (2PYO) (*Clapier et al., 2008a*) fit within the map as a rigid body using UCSF chimera. (**C**) Indicated equivalent lateral slices through free H2AK13Cub-NCP (left panels) and RNF169(UDM2) bound H2AK13Cub-NCP maps (right panels), showing the raw map density and colored according to local resolution estimates (*Kucukelbir et al., 2014*). (**D**) Histogram comparison of $S^2\tau_C$ values obtained for ILV-methyl labeled ubiquitin in free (blue) and RNF169(UDM2) bound H2AK13Cub-NCPs (pink) fit to a normal distribution.

The following figure supplement is available for figure 7:

**Figure supplement 1.** Cryo-EM data and processing.

members, Y697 and L699 sample positions which direct their sidechains toward a hydrophobic pocket formed by H2A Y49, V53 and Y56, much like the positions of M6 and L8 in the LANA peptide – NCP structure (*Barbera et al., 2006*). In addition, there are alternative conformations of L699 where it appears to make no direct contact to the NCP surface, but instead interacts with regions of the LRM2 module itself. For example, L699 is observed to interact with M704 or A705 in some members of the ensemble.

We have carried out further simulations to verify the important role of R700 by performing a series of 50 simulated annealing cycles per replica in the absence of experimental restraints and cycling the simulation temperature from 300 K to 500 K and back to 300 K in each cycle (see

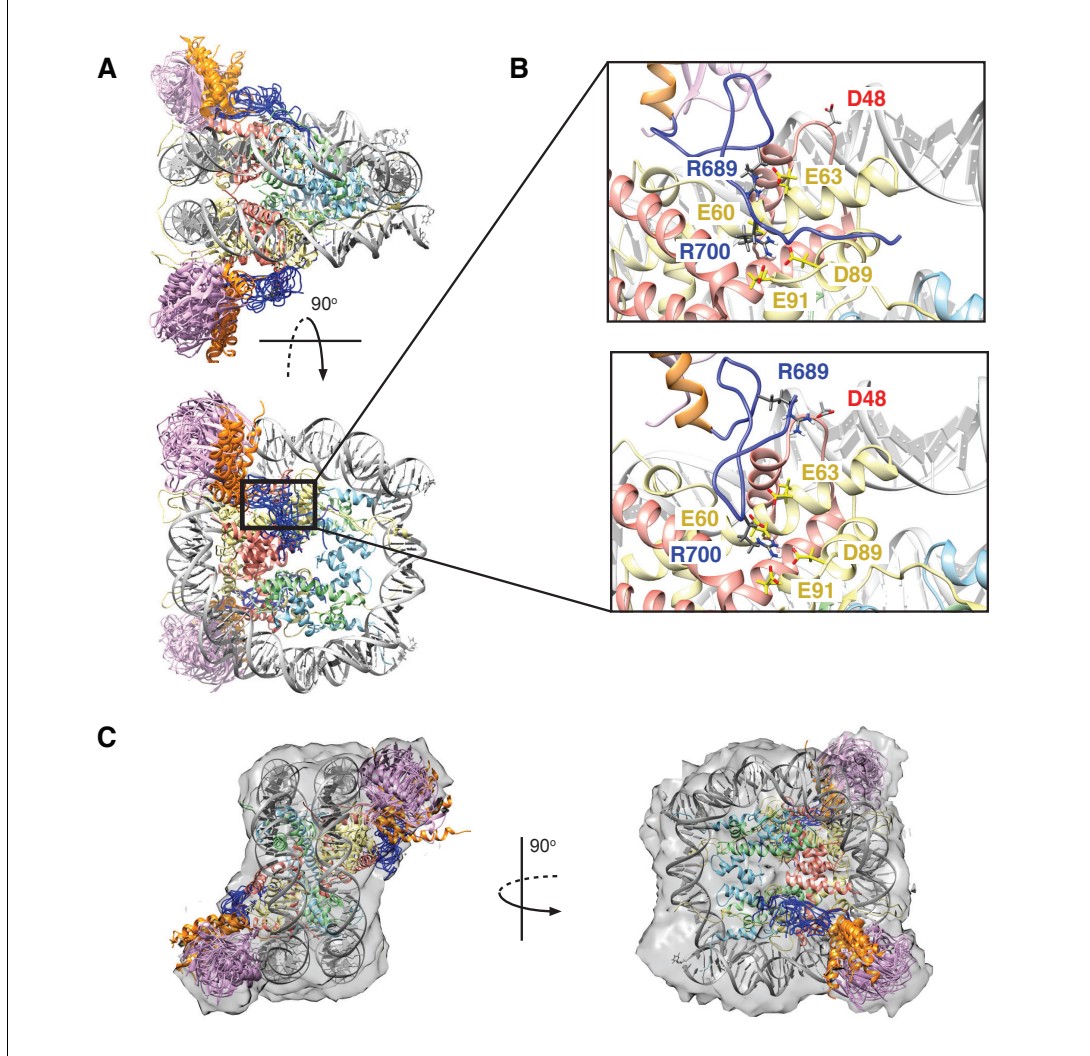

**Figure 8.** Structural model of the RNF169(UDM2)-ubNCP complex. (**A**) Alignment of ten representative members of the RNF169(UDM2)-ubNCP structural ensemble obtained from replica-averaged MD simulations constrained by CSPs and mutagenesis data. Histones H2A and H2B were used to align the structures with only one copy of the histones and DNA shown for simplicity. Ubiquitin: magenta, RNF169(662-682)(MIU2): orange, RNF169(683-708)(LRM2): blue, H2A: yellow, H2B: salmon, H3: light blue, H4: light green, DNA: gray. (**B**) Enlarged view focusing on specific contacts between R689 and R700 and the nucleosome acidic patch in two separate structures. (**C**) Two viewpoints of ten aligned members of the RNF169(UDM2) ub-NCP structural ensemble fit into the RNF169(UDM2) bound H2AK13Cub-NCP cryo-EM map. Molecular graphics images were produced using the UCSF Chimera package(*Pettersen et al., 2004*).

The following figure supplements are available for figure 8:

**Figure supplement 1.** Schematic outline of replica-averaged MD protocol used.

**Figure supplement 2.** Initial position of ub in RNF(UDM2)-ubNCP starting structures does not influence the final ub orientation.

Materials and methods). In all cases R700 remained in the canonical anchoring position in the acidic patch. As a further test to examine the possibility that R689 can replace R700 in the acidic patch we started from a structure in which the sidechain of R689 was forced into the anchoring position and subsequently carried out a series of simulated annealing cycles. After approximately eight cycles R689 moved out of the anchoring pocket and after an additional 10 cycles R700 inserted into the pocket where it remained for the final 12 cycles of the simulation.

Although the resolution of the cryo-EM model (*Figure 7A–C*) is not sufficient to obtain atomic insights, it can be used to cross-validate some aspects of our ensemble averaged restrained MD models, in particular the location of ubiquitin. In *Figure 8C*, 10 replica-averaged MD derived structures are superimposed on the cryo-EM envelope and the positions of the ubiquitin molecules (magenta) fit well to the cryo-EM map. Notably, the final position of ubiquitin does not depend on its initial position in MD simulations. *Figure 8—figure supplement 2A* highlights a pair of H2AK13Cub-NCPs with ubiquitin in different starting positions, relative to the NCP, with convergence to similar final positions obtained in both cases (*Figure 8—figure supplement 2B*).

## Discussion

We have presented a structural ensemble of the RNF169(UDM2)-H2AK13Cub-NCP complex based on replica-averaged MD simulations that included CSPs and mutagenesis data as experimental restraints. This structural ensemble illustrates how RNF169 is able to discriminate between numerous ubiquitylated chromatin species to specifically bind nucleosomes monoubiquitylated at H2AK13/K15. The multicomponent complex includes a three-pronged interaction involving a MIU2-ubiquitin component and a pair of electrostatic contacts between key arginine residues within the LRM2 module of RNF169 and the acidic patch region of the nucleosome surface. Our model reveals that the MIU2 helix interacts with ubiquitin via a hydrophobic surface centered about A673 of RNF169 (UDM2) that involves the canonical binding interface on ubiquitin, comprised of I44, L8 and V70, similar to a previously published MIU-ubiquitin crystal structure (*Penengo et al., 2006*). Notably, the LRM2 module remains highly disordered in the complex, with electrostatic contacts between R689 and one or more of H2A E60/E63 and H2B D48, and between R700 and H2A E63, D89 and E91 observed in all members of the calculated structural ensemble. The good agreement between the position of ubiquitin within the structural ensemble and in the cryo-EM map of RNF169-bound ubNCPs provides cross-validation for the position of ubiquitin in the multicomponent complex. Moreover, the cryo-EM data establishes that within the complex ubiquitin is held more rigidly to the nucleosome core, through the combined action of the MIU2 and LRM2, in agreement with NMR based spin relaxation measurements. Finally, because the critical C-terminal MIU2-LRM2 module of RNF169, that confers specificity, is conserved in RNF168 the proposed model for the RNF169 (UDM2)-H2AK13Cub-NCP complex is likely a good proxy for the interaction of K13ub-NCPs with RNF168.

The size of the RNF169(UDM2)-ubNCP complex and the intrinsic dynamics of both the histone H2A tail and the LRM2 module pose significant challenges for traditional structural biology techniques which often assume that the experimental restraints define a single conformer. The integrative approach taken here, including the use of methyl-TROSY NMR, mutagenesis, replica-averaged MD simulations and cryo-EM was essential in defining the molecular details of the complex. NMR has evolved in recent years to become a powerful tool for the study of the structure, and in particular the dynamics, of large proteins and protein complexes through the development of optimized protein labeling methods (*Gardner and Kay, 1997*; *Goto et al., 1999*; *Tugarinov and Kay, 2004*; *Kainosho et al., 2006*) and TROSY-based experimental approaches that enhance both spectral resolution and sensitivity (*Tugarinov et al., 2003*; *Fiaux et al., 2002*). Despite the prevalence of methyl bearing residues within protein cores and at interfaces in complexes (*Janin et al., 1988*), their scarcity on exposed protein surfaces and within highly charged regions can be limiting in the study of electrostatically driven interactions. Pertinent to this case is the lack of ILV residues on the surface of the NCP. While some well-positioned methyl probes, including V45 and L103 of H2B, were instrumental in identifying the acidic patch as the binding partner for the critical arginine regions of the LRM2, methyl-based NMR strategies alone were insufficient to ascertain the structural details of this complex. Distance restraints are key elements in the elucidation of detailed structures, with short (~5 Å) to long-range (< ~30 Å) distances available from NOE and paramagnetic relaxation enhancement (PRE) NMR techniques, respectively (*Battiste and Wagner, 2000*; *Wüthrich, 1986*). However, the dynamic nature of disordered protein segments can challenge the measurement of intermolecular distances (*Mittag and Forman-Kay, 2007*). In our hands, intermolecular methyl NOEs between RNF169(UDM2) and NCP histones were difficult to observe, reflecting both the dilute samples used (100 μM) and the paucity of methyl groups at the binding interface. Furthermore, in dynamic systems, such as the one studied here, intermolecular PREs provide at best upper distance bounds that

we have found to be too large to be useful for straightforward protein-protein docking approaches. To circumvent these difficulties replica-averaged MD simulations were used to account for the inherent flexibility of both the ubiquitin attachment and the LRM2 module so as to produce a structural ensemble consistent with our experimental data. Improvements in both the accuracy of the force fields that describe molecular interactions and the available computational power have increased the robustness of MD simulations and the complexity of the systems amenable to them (*Vendruscolo and Dobson, 2011*; *Chung et al., 2015*). The inclusion of experimental NMR data in MD simulations, as ensemble-averaged restraints, provides a means to more extensively sample the conformational space of interest and to generate ensembles of structures consistent with experimental measurements (*Torda et al., 1989*; *Lindorff-Larsen et al., 2005*; *Cavalli et al., 2013b*). The complementarity of the techniques used here and the continuous effort toward improving their capabilities will facilitate future studies of dynamic macromolecules, similar to the RNF169-ubNCP complex described here.

The present study reveals the molecular details of the interaction between a new class of ubiquitin reader and ubiquitylated NCPs. Recently the cryo-EM derived structure of a complex of a ubiquitylated histone reader, 53BP1, and H2AK15 ubiquitylated-NCP was reported, revealing the basis for a separate class of interaction (*Wilson et al., 2016*). In the case of 53BP1, the interaction is established via the involvement of a pair of arginine residues flanking the ubiquitin conjugation position in the H2A N-terminal tail, R11 and R17, that straddle the DNA to optimally position ubiquitin for contact with the ubiquitylation dependent recruitment motif (UDR) of 53BP1 (*Wilson et al., 2016*). Modeling of the UDR region also revealed contact with the acidic patch via a single arginine residue (R1627) (*Wilson et al., 2016*). The acidic patch surface is a critical component of all chromatin factors bound to the NCP (*Barbera et al., 2006*; *McGinty et al., 2014*; *Makde et al., 2010*; *Armache et al., 2011*; *Morgan et al., 2016*; *Kato et al., 2013*), where, at a minimum, one anchoring arginine binds a cavity generated by H2A E60, D89 and E91 side chains. In the case of RNF169, R700 acts as the anchoring arginine, with R689 making contacts of approximately equal importance to the overall stability of the complex. While the topology of the acidic patch can accommodate interactions with a variety of structural types, RNF169, similar to HMGN2 and CENP-C (*Kato et al., 2011a*, *Kato et al., 2013*), lacks defined secondary structure when bound to the NCP surface.

Ubiquitylation of H2A on K13 and K15 by RNF168 is necessary for the recruitment of downstream repair factors such as RNF169, 53BP1 and BRCA1 (*Jackson and Durocher, 2013*). The ability of RNF168 to specifically bind its own mark provides both a mechanism for facilitating the amplification of these marks as well as spatial and temporal control over its catalytic activity, ensuring the correct, stepwise execution of the DSB response. Notwithstanding the similarities of the UDM2 domains of RNF168 and RNF169, the affinity of RNF169 for K13 or K15 ubNCPs is higher (*Figure 1—figure supplement 1*). This may provide an additional level of control on the catalytic activity of RNF168 since as RNF169 accumulates at DSBs, in an RNF168-dependant manner, it can outcompete RNF168 for H2AK13/K15 ubiquitylated NCPs to maintain the ubiquitin signal within a defined region of chromatin surrounding DSBs. A complete understanding of the interplay between the RNF168/169 ubiquitin readers and downstream repair factors 53BP1 and BRCA1, and related implications for the cellular response to DNA damage, are important outstanding questions. In this report, we have presented an ensemble of RNF169(UDM2)-H2AK13Cub-NCP structures illustrating the inherent dynamics of the complex, while revealing the residue-specific contacts that impart selectivity. Our model shows how relatively weak interactions work synergistically to enable the selection of a specific type of ubNCP among the diverse array of ubiquitylated chromatin sites in the nucleus.

# Materials and methods

## Protein expression and purification

### For pull-downs and cell assays

All MBP and GFP fusion proteins as well as human and xenopus laevis histone genes were prepared as previously described (*Panier et al., 2012*; *Fradet-Turcotte et al., 2013*). Pull-down experiments (*Figure 1C and D*, *Figure 1—figure supplement 1* and *Figure 3—figure supplement 1C*) made use of various MBP fusion proteins including MBP-RNF168(110-201; DDp2222), MBP-RNF168(374-571; DDp2220), MBP-RNF169(662-708; DDp1675), MBP-RAP80(60-124; DDp1708), and MBP-RAD18

(201-240; DDp1698). Human histones genes used to prepare NCPs included pET15b His-H2A (DDp1872), pET15b His-H2B (DDp1873), and from Xenopus laevix pET3d H3 (DDp1874) and pET3a H4 (DDp1875). E2 and E3 enzyme constructs used for catalytic ubiquitylation (*Figure 1C and D*, *Figure 1—figure supplement 1* and *Figure 3—figure supplement 1C*) included pPROEX GST-RNF168 residues 1–113 (DDp1878), His6-UBCH5a (DDp1543), pET24b BMI1-His6 residues 1–108 (DDp1886) and pGEX-6P-1 RING1b residues 1–116 (DDp1887). The GFP-RNF169 used for transfection in cell assays included residues 662–708 (DDp1674) (*Figure 5—figure supplement 1*).

GST and MBP fusion proteins were produced as previously described (*Panier et al., 2012*; *Fradet-Turcotte et al., 2013*; *Juang et al., 2012*). Briefly, MBP and GST proteins expressed in *Escherichia coli* were purified on amylose (New England Biolabs) or glutathione sepharose 4B (GE Healthcare) resins according to the batch method described by the manufacturer and stored in 50 mM HEPES pH 7.5, 150 mM NaCl, 5% glycerol or in PBS. All protein concentrations were determined via the Pierce BCA assay kit (ThermoFisher), followed by SDS-PAGE Coomassie staining with comparison to known control proteins.

## For NMR and cryo-EM studies

*D. melanogaster* histone genes for H2A, H2B, H3 and H4 were cloned into a pET21b expression vector, as described previously (*Kato et al., 2011b*). The *H. sapiens* ubiquitin gene was cloned into a pET28b expression vector which included a thrombin cleavage site replaced with a N-terminal His6-tag and a tobacco etch virus (TEV) cleavage site between the $His_6$-tag and the protein sequence. The *h. sapiens* RNF169(UDM2) gene, encompassing residues 662–708, was cloned into a pET29b+ expression vector that included an N-terminal $His_6$-SUMO fusion protein removable using SUMO protease Ulp1 (*Malakhov et al., 2004*). RNF169(UDM2) including the DAAA N-terminal helix extension (that increases temperature stability) was purchased from GenScript and sub-cloned into pET29b+. It will be explicitly assumed below that RNF169(UDM2) includes the DAAA extension. All mutants including, H3(C110S), Ub(G76C), H2A(K13C), H2A(E60A/E63A), H2A(D89A/E91A), RNF169 (R689A), RNF169(Y697A), RNF169(L699I), RNF169(R700A), RNF169(S701A), RNF169(R689A/R690A/ K691A) and RNF169(L699A/R700A/S701A), were prepared with PfuTurbo DNA polymerase using the QuikChange site-directed mutagenesis method (Strategene). Protein expression was achieved by growing *Escherichia coli* cells at 37°C, transformed with the desired expression plasmid, in media containing 100 mg/L ampicillin for all histones and 50 mg/L kanamycin for ubiquitin and RNF169 (UDM2). Expression was induced with 1 mM IPTG for 16 hr at 37°C for H2A, H2B and H3, 4 hr at 37°C for H4, 16 hr at 30°C for ubiquitin and 16 hr at 18°C for RNF169(UDM2). Cells were subsequently harvested by centrifugation and purified immediately or stored at −80°C. Unlabeled proteins were expressed in LB media. Protonated, uniformly $^{13}C$, $^{15}N$-labeled RNF169(UDM2) was grown in 100% $H_2O$ minimal M9 media supplemented with 3 g/L [$^1H$-$^{13}C$]-glucose and 1 g/L $^{15}N$-$NH_4Cl$. Deuterated histones, ubiquitin and RNF169(UDM2) were expressed in 99.9% $D_2O$ minimal M9 media supplemented with 3 g/L [$^2H$-$^{12}C$]-glucose. For the expression of ubiquitin, H2A, H2B and H3C110S with Ile-$\delta$1-[$^{13}CH_3$] and Leu/Val-[$^{13}CH_3,^{12}CD_3$] methyl labeling (ILV-methyl labeling) (*Tugarinov and Kay, 2004*), 99.9% $D_2O$ minimal M9 media was supplemented with 3 g/L [$^2H$-$^{12}C$]-glucose, 85 mg/L $\alpha$-ketoisovaleric acid (L/V labeling) and 45 mg/L $\alpha$-ketobutyric acid (Ile-labeling) 1 hr prior to induction. All histones were purified from inclusion bodies using HighTrap SP-XL ion exchange columns (GE Healthcare) equilibrated in 7 M urea, 150 mM NaCl, 50 mM Tris pH 8 and eluted using a linear gradient up to 1 M NaCl over 9 column volumes. Fractions containing the histone proteins were then pooled, concentrated and further purified using a HighLoad 10/300 S75 superdex gel filtration column (GE Healthcare), followed by extensive dialysis into water and lyophilization. Ubiquitin and RNF169 were purified using Talon metal affinity resin (Clontech), followed by TEV cleavage to remove the $His_6$-tag (ubiquitin) or Ulp1 cleavage to remove SUMO (RNF169) and a final gel filtration run using a HighLoad 10/300 S75 superdex column equilibrated in 150 mM NaCl, 20 mM Tris pH 7.5 (ubiquitin) or 150 mM NaCl, 20 mM sodium phosphate pH 6 (RNF169). Protein purity was evaluated using SDS-PAGE and electrospray ionization-mass spectrometry (ESI-MS).

## Peptides

LANA1-23 peptide, (Biotin-LC-MAPPGMRLRSGRSTGAPLTRGSY) and the non-binding LANA1-23 LRS mutant peptide, (Biotin-LC-MAPPGMRAAAGRSTGAPLTRGSY) (*Figure 1—figure supplement 1*) were synthesized by BioBasic.

## DNA preparation

153 bp 601 DNA (*Lowary and Widom, 1998*) was prepared from a 32-copy plasmid (a gift from Dr. Tom Muir's lab), transformed into DH5$\alpha$ *Escherichia coli* cells and grown overnight in LB/ampicillin media at 37°C. Following harvest by centrifugation, Giga prep plasmid purification kits (Qiagen, cat.12191) were used to purify the 32-copy plasmid. Digestion using EcoRV (0.5units/$\mu$g plasmid) in NEB buffer3 was carried out for 20 hr at 37°C. The resulting solution of liberated 153 bp fragments was treated with 0.3375 vol equivalents of fresh 40% PEG-600 and 0.15 vol equivalents of 5 M NaCl solution and incubated at 4°C for 1 hr to precipitate the vector backbone. Following centrifugation at 14,000 rpm for 30 min at 4°C, 2.5 vol equivalents of 100% ethanol was added to the supernatant and left at −20°C overnight to precipitate the 153 bp 601 DNA fragments. Following centrifugation at 14,000 rpm for 30 min at 4°C, the precipitated DNA pellet was washed once with cold 70% ethanol and resuspended in 10 mM Tris pH 8. The 153 bp 601 DNA fragments were further purified using a HiTrap DEAE-FF column (GE Healthcare) equilibrated in 10 mM Tris pH 8 and eluted using a linear gradient up to 1M KCl over 9 column volumes. Fractions containing DNA were pooled, concentrated and the concentration of KCl was adjusted to 2 M for subsequent NCP reconstitution.

## Disulfide-directed ubiquitylation of H2A

Purified ubiquitin was prepared for conjugation by the addition of fresh 1,4-dithiothreitol (DTT) at 5 mM concentration to ensure all intermolecular disulfide bonds were reduced. Subsequent removal of DTT was achieved using PD-10 desalting cartridges (GE Healthcare). The eluate was flash frozen with liquid nitrogen and subsequently lyophilized. Lyophilized H2AK13C (~5 mg) was resuspended in 1 mL water with 5 mM tris(2-carboxyethyl)phosphine (TCEP) and subsequently activated by the addition of 10-fold molar excess of 2,2'-dithiobis(5-nitropyridine) (DTNP) dissolved in 2 mL of acetic acid. The reaction was allowed to proceed overnight at room temperature and verified by ESI-MS. The activated product (H2A K13C-DTNP) was dialyzed extensively into water to remove unused DTNP, followed by gel filtration using a HighLoad 10/300 superdex S75 gel filtration column (GE Healthcare) equilibrated with 6 M guanidine hydrochloride, 150 mM NaCl, 50 mM TRIS pH 6.9. Following degassing of the H2A K13C-DTNP solution lyophilized UbG76C was added at a 2:1 molar excess to the histone solution and gently agitated for 1 hr to allow for the completion of the reaction. The final product was verified with ESI-MS. The procedure is outlined schematically in *Figure 3—figure supplement 1A*.

## Octamer refolding and NCP reconstitution

Purified H2B, H3C110S, H4 and H2A K13C-Ub were combined at equimolar ratios and refolded into octamers, that were subsequently purified and reconstituted into NCPs as previously described (*Dyer et al., 2004*) with the exception that $\beta$-mercaptoethanol was not added to any buffers. Microscale test NCP reconstitution reactions (50 µL, 7 µM DNA) were used to determine the optimal stoichiometry of Ub-octamers and DNA. The quality and purity of the resulting NCPs were checked using 5% native PAGE and ESI-MS.

## Catalytic ubiquitylation of NCPs

Nucleosomes were ubiquitinated by incubating 2.5 µg recombinant mononucleosomes with 30 nM E1 (Uba1), 1.4 mM UBCH5a, 4 mM RNF168 (1–113) (*Figure 1C and D*, *Figure 1—figure supplement 1*, *Figure 3—figure supplement 1C*) or BMI1–RING1B complex (*Figure 1D*), 11 mM ubiquitin (Boston Biochem) and 4 mM ATP in a buffer containing 50 mM Tris-HCl, pH7.5, 100 mM NaCl, 10 mM MgCl$_2$, 1 mM ZnOAc and 1 mM DTT at 30°C for 2 hr.

## Antibodies

For Western blotting shown in *Figure 1C and D*, *Figure 1—figure supplement 1* and *Figure 3—figure supplement 1C* we used the following primary antibodies: rabbit anti-H3 (Ab1791, Abcam –

RRID:AB_302613), mouse anti-MBP (E8032, NEB – RRID:AB_1559732), rabbit anti-H2A (raised against amino acid residues 100–130)(*Fradet-Turcotte et al., 2013*), and rabbit anti-H2A (targeting 719 the acidic patch, 07–146, Millipore – RIDD:AB_310394). Peroxidase-affiniPure goat anti-rabbit IgG (111 035 144, Jackson Immuno Research – RRID:AB_2307391) and HRP-linked sheep anti-mouse IgG (NA931, GE Healthcare – RIDD:AB_772210) were used as secondary antibodies. For immunofluorescence and FACS analyses, as shown in *Figure 5—figure supplement 2*, cells were stained for mouse anti-γ-H2AX (clone JBW301, Millipore– RIDD:AB_309864) and rabbit anti-53-BP1 (sc-22760, Santa Cruz – RIDD:AB_2256326). The following antibodies were used as secondary antibodies in immunofluorescence microscopy: Alexa Fluor 555 anti-rabbit and AlexaFluor 647 goat anti-mouse (Molecular Probes – RIDD:AB_141784 and RIDD:AB_141725, respectively). DNA was counterstained with DAPI to trace the outline of nuclei.

## Cell culture and plasmid transfection

Human cell culture media were supplemented with 10% fetal bovine serum (FBS) and maintained at 37°C and at a 5% $CO_2$ atmosphere. U-2-OS (U2OS, RRID:CVCL_0042) WT were purchased from ATCC and cultured in McCoy's medium (Gibco). The cell line was tested to be negative for mycoplasma contamination and authenticated by STR DNA profiling. Plasmid transfections were carried out using Lipofectamine 2000 Transfection Reagent (Invitrogen) (*Figure 5—figure supplement 1*).

## Immunofluorescence microscopy

Cells were grown on coverslips, irradiated with 2Gy and fixed with 2% (w/v) paraformaldehyde in PBS 1 hr post-irradiation. Cells were then processed for immunostaining as described previously (*Panier et al., 2012*; *Escribano-Díaz et al., 2013*; *Orthwein et al., 2015*). Confocal images were taken using a Zeiss LSM780 laser-scanning microscope and a Leica SP5-II confocal microscope in standard scanning mode. (*Figure 5—figure supplement 1*)

## NCP pull-down assays

NCP pull-downs (*Figure 1C and D*, *Figure 1—figure supplement 1* and *Figure 3—figure supplement 1C*) were performed in a total volume of 100 μL by using 15–20 μL ubiquitination reaction (see Catalytic ubiquitylation of NCPs above), 2, 4 or 8 μg MBP-protein coupled to amylose resin (New England Biolabs), in pull-down buffer (50 mM Tris-Cl pH 7.5, 150 mM NaCl, 1 mM DTT, 0.05% NP-40, 0.1% BSA). Pull-down reactions were incubated for 2 hr at 4°C. Pull-downs were then washed three times with 0.75 mL of the pull-down buffer plus 0.1% BSA and eluted in Laemmli SDS–PAGE sample buffer for analysis by immunoblotting. Pull-downs presented in *Figure 3—figure supplement 2B* were eluted in Laemmli SDS–PAGE without DTT to preserve the integrity of chemically labeled ubK13C- or ubK15C-H2A NCPs. For the competition assay (*Figure 1—figure supplement 1*) the pull-down was performed as normal, with the addition of LANA peptide, MBP-RNF168(UDM2) or MBP-RNF169(UDM2) during the incubation with ubiquitylated NCP.

## Circular dichroism spectroscopy

Circular dichroism (CD) spectra and temperature melts of RNF169(UDM2) without and with the DAAA N-terminal helix extension (*Figure 2—figure supplement 1B*) were acquired using a Jasco J-815 CD Spectrometer (Jasco, Inc.) and a 0.1 cm path length cuvette. CD spectra were collected as an average of 3 scans between 190 and 240 nm using a 20 nm/min scanning rate, 8 s response time and protein concentrations in the range of 50–65 $\mu$M. For temperature melts elipticity at 222 nm was monitored over a temperature range of 10°C to 70°C with a temperature slope of 1°C/min and protein concentrations were in the 25–35 $\mu$M range. Raw data sets were corrected for buffer contributions and converted to percent helicity as previously described (*Sommese et al., 2010*; *Chen et al., 1974*).

## NMR experiments

All NMR experiments on NCPs were performed at 45°C using 14.0 T Varian Inova or Bruker Avance III HD spectrometers equipped with cryogenically cooled pulse-field gradient triple-resonance probes. The assignment of RNF169(UDM2) and titrations of free ubiquitin with RNF169(UDM2) were performed at 35°C and 45°C, respectively, on a 11.7 T Varian Inova spectrometer with a room

temperature pulse-field gradient triple-resonance probe. The NMR buffer for H2A K13Cub-NCP samples contained 100 mM NaCl, 0.05% azide, 0.5% trifluoroethanol and 20 mM sodium phosphate pH 6, 99.9% $D_2O$, with sample concentrations in the range of 50–100 uM in NCP, as determined by A260 measurement of DNA. The NMR buffer used for the assignment of RNF169(UDM2) contained 100 mM NaCl, 0.05% azide, and 20 mM sodium phosphate pH 6, 90% $H_2O$/10% $D_2O$ with 0.62 mM protein concentration. All NMR data were processed and analyzed using the suite of programs provided in NMRPipe/NMRDraw and NMRviewJ software packages (*Delaglio et al., 1995*; *Johnson and Blevins, 1994*), with backbone assignments carried out using CCPnmr (*Vranken et al., 2005*).

Translational diffusion coefficients (*Figure 3—figure supplement 1B*) were measured by recoding a series of 1D $^{13}$C-edited spectra at 25°C using a pulse sequence analogous to an $^{15}$N-edited experiment published previously (*Choy et al., 2002*), with the $^{15}$N pulses exchanged for $^{13}$C pulses. After initial gradient encoding of the magnetization a constant-delay diffusion element of 150 ms for free ubiquitin (8.9 kDa) or 200 ms for ¼ proteasome (180 kDa) and H2AK13Cub-NCP (235 kDa) was employed. The resulting $^1$H methyl signal was integrated to quantify intensities as a function of gradient strength. Diffusion constants were obtained by nonlinear least square fits of peak intensities as a function of the square of the gradient strength to the relation $I = I_o \exp(-aDG^2)$ where $I$ and $I_o$ are the integrated peak intensities in the presence and absence of the gradient $G$, respectively, $D$ is the diffusion constant and $a$ is a constant comprised of experimental parameters.

Backbone resonance assignments of RNF169(UDM2) (*Figure 2*) were completed using 2D $^1$H-$^{15}$N HSQC and 3D HNCACB, HNCACO, HNCO and CBCA(CO)NH experiments (*Sattler et al., 1999*; *Cavanagh, 2007*; *Muhandiram and Kay, 1994*), with side-chain assignments obtained using 3D H(C)(CO)NH-TOCSY and (H)C(CO)NH-TOCSY experiments (*Logan et al., 1993*; *Grzesiek and Bax, 1992*). Stereospecific assignments of leucine and valine residues were achieved as previously described (*Neri et al., 1989*). Briefly, RNF169(UDM2) was prepared in 100% $H_2O$ M9 minimal media supplemented with 10% [$^1$H,$^{13}$C]-glucose/90% [$^1$H,$^{12}$C]-glucose; subsequent analysis of CT-HSQC spectra (*Vuister and Bax, 1992*; *Santoro and King, 1992*) produced assignments of leucine $\delta 1/\delta 2$ and valine $\gamma 1/\gamma 2$ resonances.

$^1$H-$^{13}$C HMQC spectra of nucleosomes were recorded exploiting a methyl-TROSY effect to obtain high quality spectra (*Tugarinov et al., 2003*; *Wand et al., 1996*). Assignments of all ILV methyl resonances within the histones and ubiquitin were transferred from those previously published (*Kato et al., 2011b*; *Wand et al., 1996*). Chemical shift perturbations were measured using $^1$H-$^{13}$C HMQC spectra of free H2AK13Cub-NCP and H2AK13Cub-NCP saturated with 2.5-fold excess RNF169(UDM2). Weighted CSPs of ILV residues were calculated according to:

$$CSP = \sqrt{\Delta\delta_{H,i}^2 + \Delta\delta_{C,i}^2 \cdot w_i} \tag{1}$$

where $\Delta\delta_i$ is the difference in chemical shift between the free and bound states (ppm) for a given isoleucine, leucine or valine resonance $i$, and the chemical shift weighting factor $w_i$ was set to $\frac{\sigma_{H,i}}{\sigma_{C,i}}$ (~0.16–0.18), where $\sigma_i$ is the standard deviation of deposited chemical shifts for isoleucine, leucine and valine methyl resonances in the Biological Magnetic Resonance Data Bank (BMRB, http://www.bmrb.wisc.edu) for atom $i$ (*Kato et al., 2011b*).

Titrations (*Figure 3A and B* and *Figure 3—figure supplement 2*) were carried out by increasing the ratios of [RNF169]/[Ub] ([RNF169]/[H2AK13Cub-NCP]) from 0 to 42 in a series of 10 $^1$H-$^{13}$C CT-HSQC experiments (from 0 to 2.6 over a series of 16 $^1$H-$^{13}$C HMQC data sets). The titrations were followed via $^{13}$C Ub that was either uniformly labeled (titration of Ub) or ILV-methyl labeled (titration of K13Cub-NCP). For the titration of $^{13}$C-labeled RNF169(UDM2) (*Figure 3—figure supplement 3*) the ratio of [ub]/[RNF169] was increased from 0 to 6 in a series of $^1$H-$^{13}$C CT-HSQC experiments at 35°C. Note that there are two equivalents of Ub for each NCP and we have assumed independent binding of RNF169 to each Ub site. $K_D$ values were extracted from nonlinear least square fits of the resulting binding isotherms that were obtained by extracting chemical shifts in either $^{13}$C or $^1$H dimensions of $^1$H-$^{13}$C correlation plots via:

$$\Delta\delta' = \Delta\delta'_{MAX} \frac{[L]_T + [P]_T + K_d - \sqrt{([L]_T + [P]_T + K_d)^2 - 4[P]_T[L]_T}}{2[P]_T} \tag{2}$$

where $[P]_T$ and $[L]_T$ are the total concentration of Ub and RNF169, respectively, $\Delta\delta'$ is the chemical shift change (relative to the RNF169-free spectrum) at each titration point, and $\Delta\delta'_{MAX}$ is the difference between free and bound chemical shifts. Positions of individual peaks in fast exchange on the NMR chemical shift timescale were fit and reported errors in $K_D$ correspond to one standard deviation of these values.

Kinetic parameters for the RNF169 + H2AK13Cub-NCP binding reaction (*Figure 3C*) were obtained via line-shape analysis using a $K_D$ value of 24 ± 7 μM obtained from chemical shift titration data. Experimental line-shapes ($^{13}$C dimension) were extracted for I44δ1 of ubiquitin for a range of [RNF169]/[H2AK13Cub-NCP] values and fit using scripts written in MATLAB (MathWorks Inc.), as previously described (*Tugarinov and Kay, 2003*). Intrinsic transverse relaxation rates were estimated from linewidths in $^1$H-$^{13}$C HMQC spectra and were not used as fitting parameters. Peak intensities for each titration point (each [RNF169]/[H2AK13Cub-NCP] ratio) were scaled to account for differential line broadening in the $^1$H dimension that can affect intensities of extracted $^{13}$C traces. Fitted parameters included $k_{on}$ and an adjustable coefficient for each titration point, as described above.

ILV-methyl group dynamics (*Figure 7D*) have been measured for ubiquitin in H2AK13Cub-NCP whereby the build-up of methyl $^1$H triple quantum coherence is quantified as described previously (*Sun et al., 2011*). Data sets were recorded in the presence and absence of RNF169(UDM2), at 45°C. Relaxation delay values of 1, 2, 3, 4, 5, 6, 7, 8, 9, 10, 12 and 14 ms were used and peak intensities corresponding to the build-up of triple quantum coherences ($I_a$) and the evolution of single quantum coherences ($I_b$) quantified from 2D spectra at each relaxation delay. Values of the product of the square of the methyl axis order parameter, $S^2$, and the tumbling time of the complex, $\tau_C$, ($S^2\tau_C$) were extracted from nonlinear least square fits of build up curves of $I_a/I_b$ as a function of relaxation delay (*Sun et al., 2011*).

Distances connecting ubiquitin and RNF169 in an RNF169(UDM2) – ubiquitin complex were obtained by recording a $^1$H-$^1$H NOESY data set (mixing time of 200 ms, 20°C) as a first step in generating the structure of the RNF169(UDM2) - H2AK13Cub-NCP complex (*Figure 6B*). The sample was ILV-methyl labeled in ubiquitin and uniformly $^{13}$C labeled in RNF169(UDM2) and was prepared using an approximate 8-fold excess of RNF169 over ubiquitin. NOEs between A673 of RNF169 and ubiquitin are particularly important as A673 has been shown to be a central player in the interaction in a crystal structure of the Rabex MIU-ub complex (*Penengo et al., 2006*).

## Electron cryo-microscopy

Holey EM grids were prepared by nanofabrication with arrays of 500–800 nm holes as described (*Marr et al., 2014*), with the alteration of gold evaporation onto the grids as a specimen support, rather than carbon (*Russo and Passmore, 2014*). For initial screening H2AK13Cub-NCP and RNF169 (UDM2) were incubated at a 1:2.5 molar ratio in 10 mM Tris-Cl pH 7.5, 200 mM KCl, 1 mM EDTA and differentially PEG precipitated as described (*Wilson et al., 2016*); the sample was diluted to 50 mM KCl immediately prior to grid preparation. The low-salt complexes were applied to grids and allowed to equilibrate for 5 s in a FEI Vitrobot and blotted for 10 s prior to freezing in a liquid ethane/propane mixture (1:1 v/v). Grids were stored in liquid nitrogen, prior to transfer to a Gatan 626 cryotransfer specimen holder. Samples were imaged with a FEI F20 electron microscope, equipped with a field emission gun and operating at 200 kV. Movies were acquired manually in counting mode with a Gatan K2 Summit direct detector device camera using a calibrated magnification of 34,483×, resulting in a physical pixel size corresponding to 1.45 Å. During movie acquisition the sample was exposed to 1.2 electrons/Å$^2$/frame and a total exposure of 36 electrons/Å$^2$ on the specimen.

Image acquisition and data analysis revealed that only a subset of RNF169(UDM2) was bound to H2AK13Cub-NCP under the preliminary screening conditions. 175 movies were acquired for this preliminary dataset. The formation of RNF169(UDM2)-ubNCP complexes was optimized by incubating a 1:5 molar ratio of H2AK13Cub-NCP to RNF169(UDM2) in 10 mM tris pH 6.8, 30 mM KCl and 1 mM EDTA followed by differential PEG precipitation as described (*Wilson et al., 2016*), with the exception that a higher final PEG-4000 concentration of 9% (v/v) was used. A total of 867 movies were acquired for this sample consisting of RNF169(UDM2)-bound ubNCPs.

Individual frames in a movie stack were aligned and averaged using the programs alignframes_lmbfgs and shiftframes (*Rubinstein and Brubaker, 2015*). The contrast transfer function (CTF) was calculated from the averaged frames using CTFFIND4 (*Rohou and Grigorieff, 2015*). Manual

inspection of micrographs and their corresponding power spectra was performed in Relion 1.3 (*Zhao et al., 2015*). Poor micrographs with ice contamination were discarded. Particle selection, based on manually selected templates, was performed in Relion 1.3. A total of 42,343 and 301,275 particle images were selected for the ubNCP and RNF169(UDM2)-ubNCP maps, respectively. After particle image extraction, beam induced particle motion between frames was corrected with align-parts_lmbfgs (*Rubinstein and Brubaker, 2015*). A previously measured magnification anisotropy from the Toronto F20 electron microscope was corrected (*Zhao et al., 2015*). Extracted particle images were subject to 2D classification in Relion 1.3 and high-resolution class averages were selected for 3D classification (*Figure 7—figure supplement 1B*). A low pass filtered model of NCP based on PDB: 1KX5 (*Davey et al., 2002*) was used as an initial template for 3D classification. For the preliminary dataset of free ubNCP, 3D classification was performed using four classes. One class, populated with 60% of the total particles (11,063 particles) was refined further to yield the 8.1 Å H2AK13Cub-NCP structure (*Figure 7A and C*). For the dataset of RNF169(UDM2)-bound NCPs, 3D classification was performed with five classes. Particle images from one class populated with 35% of the total particles (31,760 particles) showed high-resolution features and was refined further (*Figure 7—figure supplement 1C*). The RNF169(UDM2)-ubNCP refined map was sharpened in Relion 1.3 with a B-factor of −200. Global resolution estimates were determined using the FSC = 0.143 criterion after a gold-standard refinement (*Figure 7—figure supplement 1D and E*) (*Rosenthal and Henderson, 2003*). Local resolution was estimated with ResMap (*Kucukelbir et al., 2014*). Calculations with Relion 1.3 were performed using the Hospital for Sick Children high performance computing facility.

## Haddock docking

An atomic resolution model of the MIU2-ubiquitin complex (extending from K662 to N682 of RNF169, see *Figure 2A*) was generated by using the HADDOCK modeling program (*Figure 6C*) (*Dominguez et al., 2003*). The structure of (isolated) ubiquitin that was used in the docking procedure was taken from an X-ray model of a complex of the Rabex MIU with ubiquitin (PDB 2C7M, chain B) (*Penengo et al., 2006*). The structure of the MIU2 motif for the HADDOCK calculations was built as a homology model based on the Rabex MIU, corresponding to chain A of PDB 2C7M using the program MODELLER (*Fiser and Sali, 2003*). Since the C-terminal LRM region of RNF169 is largely disordered (see below) only the N-terminal MIU2 element was included at this stage. In order to carry out the docking study, a list of 'active' and 'passive' residues was defined as required by HADDOCK. Residues L672, A673, L676 of RNF169 and residues L8, I44, V70 of ubiquitin were defined as 'active' residues based on NMR titration results (*Figure 6A*, *Figure 3A*, *Figure 3—figure supplement 3*) and solvent accessibility criteria, while 'passive' residues were calculated using the standard procedure in HADDOCK. It is worth noting that NMR titration data indicate that several additional ubiquitin residues (I30, I36, I61, L67) have CSPs upon titration with MIU2 and these could potentially be involved in binding as well. They were not included as 'active' residues, however, since the methyl groups of these Ile/Leu probes were buried inside the hydrophobic core, with low solvent accessibilities. We employed a minimum of 20% relative solvent accessibility as a cut off for inclusion as an 'active' residue. This is reduced from the recommended value of 40% used for backbone atoms in Haddock calculations, as methyl groups are less exposed. Additional restraints, measured from a 200 ms mixing time NOE spectrum of an RNF169(UDM2) – ubiquitin complex, were used. These were added in a qualitative manner as upper bound distances of 7 Å between methyl protons of A673 (RNF169) and I44δ1, L8δ1, L8δ2 (ubiquitin). Docking was performed using a standard HADDOCK protocol, and the number of structures selected after the *it0*, *it1* and *itw* stages were 1000, 200, and 200 respectively. The final 200 structures were subject to cluster-analysis. Notably, the largest cluster has the lowest average energy and the lowest HADDOCK score in this cluster (also lowest score of all final structures in any cluster). The HADDOCK model reproduces the binding interface in the X-ray structure of the Rabex MIU-ubiquitin complex (PDB 2C7M), with the key feature that A673 of MIU2 is located within the hydrophobic binding pocket formed by L8, I44 and V70 of ubiquitin. This structure was selected for further MD simulation studies, as described below. A list of synthetic NOE restraints was constructed based on this representative structure, and these were imposed during the MD simulations in order to maintain the HADDOCK generated MIU2-ubiquitin complex. The total number of restraints was 331, which included atomic pairs in the MIU2-ubiquitin complex less than 5 Å. It is worth noting that we have repeated the calculations described above by modifying the upper distance to 10 Å between

methyl protons of A673 and neighboring protons on ubiquitin. The structures within the lowest energy cluster were the same as those obtained from the original calculation using 7 Å.

## Molecular dynamics simulations

The simulations described here, summarized schematically in *Figure 8—figure supplement 1*, were performed using the Parmbsc1 (*Ivani et al., 2016*) and Amberff99SB* (*Best and Hummer, 2009*) force fields for modeling the DNA and the protein component of the RNF169(UMD2) - H2AK13ub-NCP complex, respectively, along with the TIP3P water model (*Jorgensen et al., 1983*). All simulations were carried out using the software package GROMACS (*Pronk et al., 2013*) modified with PLUMED2, an open source library for free energy calculations (*Tribello et al., 2014*) and Almost (*Fu et al., 2014*), a plugin for NMR chemical shift restraints. A time step of 2 fs was used together with LINCS constraints (*Hess et al., 2008*). The van der Waals interactions were implemented with a cutoff of 1.2 nm and long-range electrostatic effects were treated with the particle mesh Ewald method and a cut-off of 0.9 nm (*Darden et al., 1993*).

### Starting model

An initial model for the MIU2-ubiquitin complex was created using the docking program Haddock (*van Zundert et al., 2016*), as described above, with the structure of ubiquitin given by PDB ID: 1UBQ (*Penengo et al., 2006*). Synthetic NOEs that enforced the obtained Haddock structure were used in most of the subsequent computations, as described below and illustrated in *Figure 8—figure supplement 1*. The MIU2-ubiquitin complex was covalently attached to the NCP 'in silico', via a disulfide bond linkage connecting C76 and C13 of ubiquitin and H2AK13C, respectively, using GROMACS (*Pronk et al., 2013*). This linkage mimics the chemical ligation of ubiquitin and H2A that has been used experimentally (see text). The C-terminal end of RNF169 (residues 683–708) and the N-terminal end of H2A (residues 1–11) were appended to MIU2 and H2A, respectively, using Modeller (*Fiser et al., 2000*). In this manner, an initial complex was created containing a pair of ubiquitin molecules / NCP, one attached to each H2A. The model was protonated and solvated with 88,385 water molecules and 150 mM NaCl (as in the NMR experiments) in a dodecahedron box of 1950 nm$^3$ volume. The system was first energy minimized for 10,000 steps using the steepest descent algorithm, equilibrated in the position restrained NVT ensemble at 300 K for 1 ns, and further equilibrated for 1 ns in the position restrained NPT ensemble using the Berendsen barostat (*Berendsen et al., 1984*). All simulations were carried out in the canonical ensemble by keeping the volume fixed and by thermosetting the system with the modified Berendsen thermostat (*Bussi et al., 2007*). The disordered part of RNF169, comprising the important LRM2 region, was guided toward the canonical acidic patch of the NCP by applying a restraining potential on a calculated distance $d$, according to

$$E_{wall} = \begin{cases} k(d - d_{max})^2, \ d > d_{max} \\ 0 \qquad\qquad , \ d \leq d_{max} \end{cases} \tag{3}$$

where $k$ is an energy constant set to 20 kJ/(mol nm$^2$), $d_{max}$ is a maximum distance set to 25 Å and $d$ is the calculated distance between the centers of mass of two groups of so-called active residues, generated from CSP and mutagenesis experiments as being 'points of interaction' between RNF and the NCP. Active residues from one group belong to RNF169, identified by mutation, and include R689, Y697, L699, R700 and S701. Active residues from a second group belong to the NCP and include L50, L57, E60, E63, L64, D89, E91 (H2A) and V45, L99 and L103 (H2B), established on the basis of CSPs and mutagenesis. After approximately 50 ns the restraining potential $E_{wall}$ dropped to 0 (*i.e.*, the distance between the two centers of mass was less than $d_{max}$). The simulation was then extended for an additional 25 ns with the same restraining potential (*Equation 3*). A set of 331 synthetic NOEs was used to enforce the original HADDOCK structure of the MIU2-ubiquitin complex throughout the full 75 ns of simulation. This was achieved via a potential of the form

$$E^{NOE} = \beta \sum_{i=1}^{noes} \left( d_i^{model} - d_i^{cal} \right)^2 \tag{4}$$

where the summation includes all NOEs in the synthetic set, $d_i^{model}$ and $d_i^{cal}$ are distances from the

initial HADDOCK model and calculated from the structure during the simulation, respectively, and the coefficient $\beta$ is set to 0.25 kJ/(mol nm$^2$). The extended trajectory was used to select four starting conformers (time points: 10 ns, 15 ns, 20 ns and 25 ns) for a replica-averaged simulation with four replicas (*Figure 8—figure supplement 1*). Before carrying out the replica-averaged simulation, each of the four structures was equilibrated in the position restrained NVT ensemble at 300 K for 5 ns and in the position restrained NPT ensemble for an additional 5 ns.

## Replica averaging

Replica-averaged molecular dynamics simulations (*Cavalli et al., 2013a*; *Kukic et al., 2016, 2014b*) were performed using all structural restraints as replica-averaged. These restraints included the chemical shifts measured for MIU2 in the unbound state that were applied to restrain the average value of the CamShift back-calculated NMR chemical shifts of MIU2 (*Kohlhoff et al., 2009*) using the restraining potential,

$$E^{CS} = \alpha \sum_{k=1}^{21} \sum_{l=1}^{5} \left( \delta_{kl}^{exp} - \frac{1}{M} \sum_{m=1}^{M} \delta_{klm}^{calc} \right)^2 \tag{5}$$

where $\alpha$ is the force constant, $k$, $l$ and $m$ run over 21 residues that include the MIU2 region ($k$), all five backbone nuclei for which chemical shifts have been measured ($l$: C$_\alpha$, C′, H$_\alpha$, H$_N$ and N) and M = 4 replicas ($m$), respectively. Also included are the derived NOE distances between MIU2 and ubiquitin, based on the HADDOK model of the MIU2-ubiquitin complex, that were used as replica-averaged structural restraints using the restraining potential

$$E^{NOE} = \beta \sum_{i=1}^{noes} \left( d_i^{model} - \left( \frac{1}{M} \sum_{m=1}^{M} \left( \frac{1}{r_m^6} \right) \right)^{-\frac{1}{6}} \right)^2 \tag{6}$$

where $\beta$ is the force constant and $i$ and $m$ run over the measured NOE restraints and the M = 4 replicas, respectively. It should be emphasized that the NOE potential is used to ensure that the structure of the ubiquitin/MIU2 helix, as determined by HADDOCK docking (see above), is preserved during the molecular dynamics calculations. To this end we have used a series of 331 synthetic NOEs based on the HADDOCK model that guide ubiquitin/MIU2 helix docking during molecular dynamics. While the 1/r$^6$ term in *Equation 6* does assume that averaging between replicas is slow compared to the overall tumbling of the NCP complex (~100 ns) this level of detail does not affect the resulting ubiquitin/MIU2 structures that are simply enforced to their HADDOCK model by this procedure. CSPs and mutagenesis data were taken into account in the following manner. For each active residue belonging to RNF169 the number of contacts with active residues from the NCP was calculated as

$$S_i = \sum_j s_{ij} \tag{7}$$

where $i$ and $j$ are active residues belonging to the RNF169 and to the NCP, respectively, $s_{ij}$ is 1 if a contact between any atom of residues $i$ and $j$ is formed and zero otherwise. The distance for the contact switching function was set to 6.5 Å in a first set of simulations and 10 Å in a second set; the structural features of the resulting ensembles do not change. The number of contacts $S_i$ was enforced to be larger than 1 in at least one replica at each time point of the simulation using a restraining potential of the same type as in *Equation 3*. Thus, the experimental measurements were used to modify the underlying force field on the fly by employing a replica averaging procedure whereby back-calculated parameters are compared with their experimentally measured values. In this procedure, the system evolves with a force field that is perturbed to increase the agreement with the experimental restraints.

In the first 75 ns of the replica-averaged simulation the values for the force constants $\alpha$ (*Equation 5*), $\beta$ (*Equation 6*) and $k$ (for the contact number, *Equation 3*) were gradually increased until the restraining potentials reached a plateau. The value of the force constant $\alpha$ was increased over a range extending from 0 to 24 kJ/(mol ppm$^2$) in steps of 1 kJ/(mol ppm$^2$), $\beta$ was increased in the range 0–2 kJ/(mol nm$^2$) in steps of 0.1 kJ/(mol nm$^2$) and $k$ in the range 0–50 kJ/(mol nm$^2$) in steps of 1 kJ/(mol nm$^2$). The simulation was then extended for an additional 125 ns. Only the final 50 ns of

the extended simulation were used for the analysis of the RNF169(UDM2) H2AK13ub-NCP complex by saving the NCP structure every 1 ps and NCP+water every 10 ps.

## Simulated annealing

Although the use of replica-averaged structural restraints enables one to improve the quality of the force field, it requires substantial computational resources to exhaustively sample the conformational space of a system as large as the one studied here. In this regard, we have further verified that R700 is the RNF169 anchoring arginine in our model by performing a series of annealing cycles between 300 K and 500 K using the final conformations of the four replicas (one each) as starting structures. The simulations were performed without the use of the experimental restraints and each annealing cycle lasted for 1 ns. In particular, the system was initially heated from 300 K to 500 K for 400 ps, kept at 500 K for 100 ps, cooled down to 300 K for 400 ps and then finally to 300 K for 100 ps. The total simulation time was 50 ns per replica (50 annealing cycles per replica). Subsequent analysis was performed only on structures extracted from low constant temperature frames (300 K), by saving conformers at the end of each annealing cycle. It is noteworthy that R700 always remains in the canonical anchoring position in the acidic patch.

To explore the possibility that R689, a second arginine residue that has been shown by mutagenesis to play an important role in the RNF169-NCP complex, can also serve as the 'canonical' anchoring arginine, we carried out combined restrained MD and simulated annealing simulations starting from the final conformation that was produced from one of the four simulations during the replica averaging procedure discussed above. Here the side chain of R689 was forced into the acidic patch of the NCP, lined by E60, D89 and E91 using a restrained MD simulation with a restraining potential of the same type as in *Equation 3*. The minimal number of contacts between these 3 acidic patch residues and R689 was set to 10 and the distance for the switching function ($d_{max}$ in *Equation 3*) was 6.5 Å. The restrained simulation was carried out by increasing $k$ in the range 0–80 kJ/(mol nm$^2$), in steps of 4 kJ/(mol nm$^2$). After approximately 20 ns of simulation time the restraining potential dropped to 0, indicating that R689 occupied the canonical arginine position in the acidic patch of the NCP. The final structure of this restrained simulation was then used as the starting conformer in a simulated annealing procedure identical to the one described above. The annealing was performed without the use of the experimental restraints and with a total simulation time of 30 ns (30 annealing cycles). After the first 8 annealing cycles R689 no longer occupied the canonical position and by approximately 18 ns it was replaced by R700 that remained in the canonical position until the end of the simulation.

## Acknowledgement

This work was supported by grants from the Canadian Institutes of Health Research to LEK and DD (FDN143343), by a Grant-in-Aid from the Krembil Foundation (DD) and by a Scholarship for the Next Generation of Scientists from Cancer Research Society (AF-T). JK-L was supported by a fellowship from the Leukemia and Lymphoma Society. LEK, DD, CHA, and AF-T hold Canada Research Chairs in Biochemistry, in Molecular Mechanisms of Genome Integrity, in Structural Genomics, and in Molecular Virology and Genomic Instability, respectively. The SGC is a registered charity (number 1097737) that receives funds from AbbVie, Bayer Pharma AG, Boehringer Ingelheim, Canada Foundation for Innovation, Eshelman Institute for Innovation, Genome Canada through Ontario Genomics Institute, Innovative Medicines Initiative (EU/EFPIA) [ULTRA-DD grant no. 115766], Janssen, Merck & Co., Novartis Pharma AG, Ontario Ministry of Economic Development and Innovation, Pfizer, São Paulo Research Foundation-FAPESP, Takeda, and the Wellcome Trust.

## Additional information

### Funding

| Funder | Grant reference number | Author |
|---|---|---|
| Canadian Institutes of Health Research | RN203972 - 310401 | Daniel Durocher |
| Canadian Institutes of Health | | Lewis E Kay |

Research

The funders had no role in study design, data collection and interpretation, or the decision to submit the work for publication.

## Author contributions

JK-L, Conceptualization, Data curation, Formal analysis, Investigation, Writing—original draft, Writing—review and editing; AF-T, Conceptualization, Data curation, Formal analysis, Writing—review and editing; PK, MDW, GP, TY, Data curation, Formal analysis; SP, SD, MDC, HvI, Data curation; CHA, JLR, Supervision; MV, Formal analysis, Supervision; DD, Conceptualization, Data curation, Supervision, Writing—original draft, Writing—review and editing; LEK, Conceptualization, Resources, Data curation, Formal analysis, Supervision, Writing—original draft, Writing—review and editing

## Author ORCIDs

Julianne Kitevski-LeBlanc, http://orcid.org/0000-0002-6608-1187

Amélie Fradet-Turcotte, http://orcid.org/0000-0002-5431-8650

Marcus D Wilson, http://orcid.org/0000-0001-9551-5514

Tairan Yuwen, http://orcid.org/0000-0003-3504-7995

John L Rubinstein, http://orcid.org/0000-0003-0566-2209

Michele Vendruscolo, http://orcid.org/0000-0002-3616-1610

Daniel Durocher, http://orcid.org/0000-0003-3863-8635

Lewis E Kay, http://orcid.org/0000-0002-4054-4083

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
