## [Decision Letter]

Thank you for submitting your article "RNF168 and RNF169 define a new class of ubiquitylated-histone reader involved in the response to DNA damage" for consideration by *eLife*. Your article has been reviewed by three peer reviewers, and the evaluation has been overseen by Ivan Dikic as the Senior and Reviewing Editor, who also acted as Reviewer #2. The other individuals involved in review of your submission have agreed to reveal their identity: Hashim Al-Hashimi (Reviewer #1) and Volker Dötsch (Reviewer #3).

The reviewers have discussed the reviews with one another and the Reviewing Editor has drafted this decision to help you prepare a revised submission.

Summary:

In this manuscript, Kitevski-LeBlanc and colleagues use a combination of structural methods including NMR, cryo-EM, biochemistry and molecular modeling to elucidate how Rnf169, and by extension, Rnf168, bind to ubiquitylated histone H2A in the context of nucleosome particles. These ubiquitin dependent interactions are necessary for the specific recruitment of repair factors to chromatin regions flanking double strand breaks. This study highlights the power of combining solution methyl TROSY NMR techniques to obtain information regarding dynamics in large assemblies at specific amino acid residues; with the lower resolution but broader coverage structural information from CryoEM; and experimentally restrained computational molecular dynamics simulations as well as site-directed mutagenesis provides for addressing detailed mechanistic questions in large complex, conformationally heterogeneous systems. The structural ensemble of the RNF169(UDM2)-H2AK13Cub-NCP complex generated with this approach helps to explain how through a multitude of weak interactions, RNF169 can discriminate among many different ubiquitylated chromatin sites to bind specifically to nucleosomes monoubiquitylated at H2AK13/K15.

Essential revisions:

1) The paper is so rich in details (that are also necessary to be able to understand how the model was built) that it gets sometimes hard to read. We would recommend to move some of the details (like details in the chapter "RNF169 and the LANA peptide compete for the acidic patch surface") to the supplementary material section. (Note from RS – we do not have a traditional supplementary material section because we do not believe in burying things in a less accessible location. You may wish to embed the text the reviewer refers to in a section attached to a primary figure, but the alternative would be to rewrite for greater clarity).

2) Few technical details about NMR method used: How were the distances between MIU2 and ubiquitin derived from the NOEs? Was isotropic overall tumbling assumed and is this reasonable? What is the justification for integrating the NOEs as averaged distances in the replica exchange which presumably assumes slow averaging? Especially for flexible residues, couldn't the NOEs experience contributions from faster motions which could complicate the interpretation as a simple average of 1/r6? Can the authors speculate as to why they were unable to observe intermolecular methyl NOEs between RNF169(UDM2) and NCP histones?

What was the criterion that was used for choosing CSPs to include as 'active' into HADDOCK docking? Was a particular solvent accessibility cut off used?

3) It is important to demonstrate that the C76Ub-C13H2A side-chain disulfide linkage supports Rnf169 binding by showing that MBP-Rnf169(UDM2) pulls down H2AK13Cub-NCP as efficiently as catalytically prepared H2AK13Ub-NCPs containing the isopeptide linkage. This has not yet been properly demonstrated. In fact, Figure 3—figure supplement 1 shows binding to histone H3, rather than H2A. As such, assessing binding to H3 is an indirect read out of binding to H2AK13Cub. A direct comparison of MBP-Rnf169(UDM2) binding between chemically and enzymatic H2A-K13-Ub needs to analyzed in the same gel exactly as shown in Figure 1 or Figure 1. Because H2AK13Cub is the basis for all subsequent studies, this analysis should be included.

4) The authors explicitly state and show that Rnf169(UDM2) binding to H2AK13-Ub is more robust than Rnf168(UDM2) binding to H2AK13-Ub. Since the authors did not empirically scrutinize Rnf168 in detail, the authors should not explicitly state, in the title and elsewhere, that Rnf168 binding to ubiquitylated histone H2A is analogous to Rnf169. While it is likely to be the case, given the strong sequence conservation between UDM2 amongst Rnf168 and Rnf169, it is still possible that subtle but important differences exist.

---

## [Author Response]

*Essential revisions:*

1) The paper is so rich in details (that are also necessary to be able to understand how the model was built) that it gets sometimes hard to read. We would recommend to move some of the details (like details in the chapter "RNF169 and the LANA peptide compete for the acidic patch surface") to the supplementary material section. (Note from RS – we do not have a traditional supplementary material section because we do not believe in burying things in a less accessible location. You may wish to embed the text the reviewer refers to in a section attached to a primary figure, but the alternative would be to rewrite for greater clarity).

We have removed the “RNF169 and LANA peptide compete for the acidic patch surface” section, replacing it with a short paragraph that includes only the details relevant to the development of our structural model, at the end of the preceding section entitled “Identification of key residues within the LRM2.” Figures associated with the rewritten primary text have been included as supplements 3A and 3B to primary Figure 5.

In the subsection “Identification of key residues within the LRM2”, we include:

“As mentioned above, the acidic patch is a common interaction surface for nucleosome binding proteins that typically involves key contacts between an arginine sidechain and the nucleosome surface.[…] While these results were unable to definitively identify the anchoring arginine in RNF169(UDM2), the fact that both triple mutants bound H2AK13Cub-NCPs reinforces the importance of both the RRK and LRS regions of RNF169(UDM2)”.

We removed the “RNF169 and LANA peptide compete for the acidic patch surface” section.

*2) Few technical details about NMR method used: How were the distances between MIU2 and ubiquitin derived from the NOEs? Was isotropic overall tumbling assumed and is this reasonable? What is the justification for integrating the NOEs as averaged distances in the replica exchange which presumably assumes slow averaging? Especially for flexible residues, couldn't the NOEs experience contributions from faster motions which could complicate the interpretation as a simple average of 1/r6? Can the authors speculate as to why they were unable to observe intermolecular methyl NOEs between RNF169(UDM2) and NCP histones?*

*What was the criterion that was used for choosing CSPs to include as 'active' into HADDOCK docking? Was a particular solvent accessibility cut off used?*

We address these details in the revised version of the text. The 7Å distances used in the initial set of calculations were taken to be approximate upper bounds on distances that would be expected to be observed in NOESY spectra. We have repeated the calculations using a 10Å cutoff and the results obtained from HADDOCK computations are very similar. In the subsection “Haddock Docking” of the revised manuscript we write:

“It is worth noting that we have repeated the calculations described above by modifying the upper distance to 10Å between methyl protons of A673 and neighboring protons on ubiquitin. The structures within the lowest energy cluster were the same as those obtained from the original calculation using 7Å.”

We address the 1/r^6^ averaging (that assumes that replicas exchange at a rate that is slower than the overall tumbling time of the NCP ~ 100 ns) in the subsection “Starting model” of the revised paper:

“It should be emphasized that the NOE potential is used to ensure that the structure of the ubiquitin/MIU2 helix, as determined by HADDOCK docking (see above), is preserved during the molecular dynamics calculations. To this end we have used a series of 331 synthetic NOEs based on the HADDOCK model that guide ubiquitin/MIU2 helix docking during molecular dynamics. While the 1/r^6^ term in Eq. 6 does assume that averaging between replicas is slow compared to the overall tumbling of the NCP complex (~100 ns) this level of detail does not affect the resulting ubiquitin/MIU2 structures that are simply enforced to their HADDOCK model by this procedure.”

The fact that intermolecular NOEs were not observed likely reflects the rather poor quality of NOESY spectra in the context of the NCP resulting from the low concentrations (typically 100 μM) used, the relatively poor spectral quality of the RNF169(UMD2)-ubNCP complex and the paucity of ILV methyl groups in key regions at the interface between RNF169 and the NCP. We briefly comment on this in the Discussion section of the revised text:

“In our hands intermolecular methyl NOEs between RNF169(UDM2) and NCP histones were difficult to observe, reflecting both the dilute samples used (100 μM) and the paucity of methyl groups at the binding interface.”

Finally, we have further described the criteria used to choose active residues for the HADDOCK calculations in subsection “Haddock Docking” of the revised text:

“We employed a minimum of 20% relative solvent accessibility as a cut off for inclusion as an ‘active’ residue. This is reduced from the recommended value of 40% used for backbone atoms in Haddock calculations, as methyl groups are less exposed.”

*3) It is important to demonstrate that the C76Ub-C13H2A side-chain disulfide linkage supports Rnf169 binding by showing that MBP-Rnf169(UDM2) pulls down H2AK13Cub-NCP as efficiently as catalytically prepared H2AK13Ub-NCPs containing the isopeptide linkage. This has not yet been properly demonstrated. In fact, Figure 3—figure supplement 1 shows binding to histone H3, rather than H2A. As such, assessing binding to H3 is an indirect read out of binding to H2AK13Cub. A direct comparison of MBP-Rnf169(UDM2) binding between chemically and enzymatic H2A-K13-Ub needs to analyzed in the same gel exactly as shown in Figure 1 or Figure 1. Because H2AK13Cub is the basis for all subsequent studies, this analysis should be included.*

We have replaced the pull-down in Figure 3—figure supplement 1 with one which directly monitors interactions with H2AK13C using an appropriate antibody.

*4) The authors explicitly state and show that Rnf169(UDM2) binding to H2AK13-Ub is more robust than Rnf168(UDM2) binding to H2AK13-Ub. Since the authors did not empirically scrutinize Rnf168 in detail, the authors should not explicitly state, in the title and elsewhere, that Rnf168 binding to ubiquitylated histone H2A is analogous to Rnf169. While it is likely to be the case, given the strong sequence conservation between UDM2 amongst Rnf168 and Rnf169, it is still possible that subtle but important differences exist.*

We have adjusted title in the revised manuscript to reflect the reviewers comment. We have also toned down our statements regarding RNF168 interactions. For example in the Discussion section of the revised text we now write:

“Finally, because the critical C-terminal MIU2-LRM2 module of RNF169, that confers specificity, is conserved in RNF168 the proposed model for the RNF169(UDM2)-H2AK13Cub-NCP complex is likely a good proxy for the interaction of K13ub-NCPs with RNF168.”